# Data-driven analysis and forecasting of highway traffic dynamics

A. M. Avila 🆔 1✉ & I. Mezić 1✉

The unpredictable elements involved in a vehicular traffic system, like human interaction and weather, lead to a very complicated, high-dimensional, nonlinear dynamical system. Therefore, it is difficult to develop a mathematical or artificial intelligence model that describes the time evolution of traffic systems. All the while, the ever-increasing demands on transportation systems has left traffic agencies in dire need of a robust method for analyzing and forecasting traffic. Here we demonstrate how the Koopman mode decomposition can offer a model-free, data-driven approach for analyzing and forecasting traffic dynamics. By obtaining a decomposition of data sets collected by the Federal Highway Administration and the California Department of Transportation, we are able to reconstruct observed data, distinguish any growing or decaying patterns, and obtain a hierarchy of previously identified and never before identified spatiotemporal patterns. Furthermore, it is demonstrated how this methodology can be utilized to forecast highway network conditions.

1 Department of Mechanical Engineering, University of California Santa Barbara, Santa Barbara, CA 93106, USA. ✉email: allanavila@ucsb.edu; mezic@ucsb.edu

Highway traffic congestion in 2013 cost Americans \$124 billion in direct and indirect losses[1]. This number is higher for some European countries[1] and is only expected to rise without the development of intelligent transportation systems (ITS) and the accurate forecasting of traffic conditions that ITS rely on to mitigate traffic. Traditionally, the analyzing and forecasting of highway traffic was performed via simulations of mathematical models[2,3]. However, the combination of data availability, modern processing capabilities, and development of machine learning (ML) algorithms has enabled an enormous amount of research into empirical data-driven algorithms[4–13]. The first class of data-driven algorithms were primarily parametric models that rely on the user's ability of accurately estimating the model's parameters. Historically, this point of view led to the use of linear and nonlinear regression techniques[6,9], Kalman filtering[7] and time-series models[8,10]. Recently there has been a large and growing interest in non-parametric models that rely on historical training data to estimate its own parameters. Among the most popular of these methods include the neural and deep neural network models (NN,DNN)[11,14–16], K-nearest neighbors[12,13], Bayesian networks[17], and (parametric/non-parametric) hybrid models[4]. Nevertheless, describing and forecasting the time evolution of traffic systems remains a challenging problem[2,18–24].

A majority of the published literature in this field has focused on primarily testing and validating the ability of a particular method to accurately describe real-world traffic data. However, the issue discussed much less than accuracy is that of a model or algorithm's capability of generalizing to a real-world implementation[2,21]. Many of the traditional mathematical models are known to be unfeasible for real-time implementation, tedious to solve numerically and depend on parameter accuracy[2]. The modern ML-based methods also depend heavily on accurate parameters and typically require large amounts of training data[4,5,11,21], which is usually limited and costly to collect[19,21]. Therefore, even if the state-of-the-art traffic models were accurate, typically they would require an unrealistic amount of data collection and parameter tuning to function across differing highways[5,21,25]. The first attempts at empirically characterizing the country-specific differences of highway traffic can be found in ref. [25], where traffic data from the US, UK, and Germany were empirically analyzed and compared. This international comparison was motivated by the fact that different countries have different infrastructure, vehicle class mix, driving rules, and even different driver behavior. Indeed, the works of ref. [25] confirm key differences in the periods of oscillation and speeds of propagation of traffic jams between the three countries. It is further stated how this country-specific dynamics of traffic will require the re-calibration of current models or the development of more general models. The findings of ref. [25] validate our view, in that some of the shortcomings of previous research approaches are not primarily their lack of accuracy but more so their heavy dependence on parameters and large amounts of training data. This renders many state-of-the-art techniques being developed today unfeasible for a large scale global implementation across differing highways[21].

In addition to the stochastic features of traffic, wave-like patterns have also been identified within traffic data[3,19,20,26–30]. The exact cause of such traffic waves is still an open topic, although several mechanisms have been proposed[20,31,32]. A common theme across many of these proposed mechanisms is the effect that lane changing can have on a highway system. The empirical works of Ahn[27,29] and Laval[33] provide evidence showing that lane changing maneuvers are key in the development of traffic waves. Unfortunately, research into multi-lane traffic dynamics has proven to be an even more challenging task[27,31,32,34]. The complex lane changing dynamics and human interaction that occurs within a multi-lane highway has restricted many state-of-the-art techniques to analyze and forecast traffic at the highway corridor scale and generalizing to the multi-lane scenario is often very difficult[2,31,32,34]. Ultimately, traffic management is generally applied at the network level[21]. However, an accurate and efficient method for the analysis and forecasting of multi-lane highway network conditions is perhaps the most difficult and strongly lacking component of modern ITS[2,14,22–24]. Furthermore, many state-of-the-art techniques often times require extensive parameter tuning and the proper pre-processing of raw data to perform adequately[21]. This has lead to the common practice of removing previously computed seasonal averages, aggregating and smoothing raw data[2,20,35,36]. Additionally, the differing dynamics between weekday, weekend, holiday, and adverse weather conditioned traffic has led to the common practice of utilizing case-specific training data to forecast only case-specific data[4,21,37,38]. Lastly, the challenge in forecasting multiple detector data typically results in verifying methods over only a single or possibly few detectors[4,11,30,37]. Therefore, many state-of-the-art benchmarks have been obtained at the highway corridor (single lane) level, over a limited number of sensor locations and are incapable of generalizing to handle the multi-lane network scenario without extensive re-training.

Overall, a systematic and accurate method for identifying, analyzing and forecasting spatiotemporal traffic features from data is still an open and challenging issue[2,18–20]. In this work, we demonstrate how the spectral properties of the Koopman operator, specifically the Koopman mode decomposition (KMD), can offer a model-free, parameter-free, data-driven approach for accurately identifying, analyzing and forecasting spatiotemporal traffic patterns. The methods we develop allow one to distinguish any growing or decaying phenomena and obtain a hierarchy of coherent spatiotemporal patterns hidden within the data. Furthermore, the forecasting scheme we propose readily generalizes to the much-needed scenario of multi-lane highway networks without any loss to its performance or efficiency. We do not rely on large historical training data nor do we distinguish between weekday, weekend, holiday, or adverse weather conditions. Our method's performance does not rely on parameter tuning or selection., Thereby providing a completely efficient and accurate method of analyzing and forecasting traffic patterns at the levels required by modern ITS.

## Results

**The Koopman mode decomposition.** The Koopman family of operators of a dynamical system is a group of infinite-dimensional, linear operators that describe the time evolution of an observable (measurable quantity) under the dynamics of the system[39,40]. From this viewpoint measurements and data can be used to interpret the underlying dynamics of a complex system via the spectral properties of the associated Koopman operator. The spectrum of the Koopman operator leads to a "triple decomposition" of a nonlinear and non-stationary dynamical system into its mean, periodic, growing or decaying, and fluctuating components[41–43]. Part of the discrete spectrum (eigenvalues and eigenfunctions) of this linear operator accurately describes the mean (period zero) and periodic components of a nonlinear dynamical system. The continuous spectrum (spectral measure) of the operator captures the stochastic or chaotic dynamics of the system. In systems with purely discrete spectrum the Koopman modes, corresponding to a particular choice of observable, allow one to reconstruct and forecast the observed quantity[41–44]. Together, the Koopman eigenvalues, eigenfunctions, and modes yield the Koopman mode decomposition KMD of a purely

discrete spectrum observable[41–43]. With the KMD in hand, one can decompose the observed quantity into a hierarchy of simpler, yet dynamically important, sub-patterns which describe the behavior of the complex system. Koopman modes describe the shape of dynamically important spatiotemporal patterns found within the data and the eigenvalues describe how these modes evolve (repeat, grow or decay) in time. The real part of a Koopman eigenvalue yields the growth or decay rate of a mode and, in the case of decay, is a measure of how long the pattern persists within the data. Similarly, the imaginary part of the Koopman eigenvalue is used to compute the period of oscillation of the mode and is a measure of how frequently the pattern repeats within the data. An appealing feature of the KMD is its ability to be directly computed from real-world data[41–43,45–48] via several numerical algorithms that have been developed. A large portion of these methods belong to the class of algorithms known as dynamic mode decomposition (DMD)[45,48–54]. In this work, we have utilized the Hankel dynamic mode decomposition algorithm (Hankel-DMD)[49,55] to approximate the Koopman modes and eigenvalues of traffic data. In what follows we demonstrate how the KMD can decompose the Next Generation Simulation (NGSIM) traffic data into dynamically important sub-patterns, hidden within the data, and identify their temporal characteristics.

**Koopman mode analysis of spatiotemporal traffic data.** We begin by studying the NGSIM data set collected by the US Federal Highway Administration. The NGSIM data set provides the precise location of every vehicle, its lane position and location relative to other vehicles for every one-tenth of a second on 2100 and 1640 ft segments of the southbound US-101 and eastbound I-80 highways, respectively. Overall, the NGSIM data provides a microscopic description of traffic in that, it is the individual vehicles that are tracked and not the velocity or density of the bulk, macroscopic flow. However, in this work, we are interested in identifying macroscopic spatiotemporal patterns and therefore the NGSIM trajectory data is converted into spatiotemporal coarse grained data via the binning method developed in ref. [56] and utilized by the authors in refs. [57,58]. This procedure allows the construction of macroscopic velocity and density profiles from vehicle trajectory data. The resulting spatiotemporal data is a matrix whose columns correspond to time, its rows correspond to a position along the highway and the entries contain the velocity, density or flow at that location and time. The resulting spatiotemporal data for the US-101 highway is shown in Fig. 1, and the I-80 highway data can be referenced in Supplementary Figs. 1 and 2. A more detailed discussion on the binning method and formulas can be referenced in the Methods section and access to the spatiotemporal data is also made available.

In this work, we categorize traffic patterns according to ref. [20]. In addition to the well-known free-flowing and congested traffic states some of the various patterns identified in ref. [20] are the pinned localized cluster (PLC), moving localized cluster (MLC), stop and go waves (SGW), and oscillating congested traffic (OCT). PLC type traffic oscillations do not propagate along the highway but are instead pinned or localized at a certain spatial location. On the other hand, MLC type phenomena, also called traffic jams, do propagate backward along the highway, affect the entire highway and their amplitudes are not perturbed by on or off-ramps. The SGW and the OCT according to ref. [20] are almost indistinguishable without the proper data filtering technique and thus, in this work, we simply refer to both as SGW or traffic waves. The presence of such patterns for the US-101 highway data can be seen in Fig. 1a.

By applying a KMD to the velocity data in Fig. 1a, we seek to uncover traffic patterns that may be hidden within the data. Patterns uncovered by the KMD for the US-101 highway density and flow as well as for the I-80 highway can be referenced in Supplementary Figs. 4–12. In our works, we sort the resulting Koopman modes according to their period of oscillation. Therefore, the slowest evolving pattern is what we refer to as the "first" mode and so on. A table listing the exact periods of oscillation of the modes discussed can be referenced in Supplementary Table 1. By plotting some of the leading Koopman modes in Figs. 2 and 3, we find, that the first three modes (Fig. 2a–c), modes 5 (Fig. 2e), 10 (Fig. 3d), 11, 18, 19 (Supplementary Fig. 3a, d, e), and 13 (Fig. 2e) all share the common structure of a PLC. Specifically, their amplitude is entirely localized around the post-off-ramp (1280–2100 ft) section of the highway. However, mode 5 is also spatially localized about the mid-ramp section of the highway. The double-peaked structure of mode 5 strongly resembles a sort of spatial harmonic feature of modes 1–3. Interestingly, mode 5 along with modes 8 (Fig. 3b), 9 (Fig. 3c), and 13 (Fig. 3e) display a standing wave node at precisely the on and off-ramp locations, which have been labeled with dark orange dotted lines. Modes 16 (Fig. 3f), 20, 21, 25, and 28 (Supplementary Figs. 3f and 4a, d, f) differ from the other PLC waves in that their amplitudes appear to grow or decay in time. The ability to uncover such growing and decaying patterns is a strongly distinguishing feature between the KMD and a Fourier analysis. It is also clear to see how the amplitudes of modes 7 (Fig. 3a) and 26 (Supplementary Fig. 4e) are unperturbed as they travel along the highway, indicating that they correspond to traffic jams that affect the entire highway as they propagate by. Several modes within Figs. 2 and 3 are harmonics of the first mode, which corresponds well with the known fact that harmonics of eigenvalues are also eigenvalues of the Koopman operator[41–43]. A complete list containing the periods of oscillation of the modes we discussed can be referenced in Supplementary Table 1.

We now demonstrate how the patterns we identify relate to previous research efforts and offer new insight. The empirical findings of Ahn[26] demonstrate how the amplitude of a traffic wave decreases when it propagates upstream past an on-ramp. Similarly, it was postulated that the amplitude should increase when propagating past an off-ramp. However, no validation for the off-ramp scenario is found in refs. [25,26]. This phenomenon was referred to as the "pumping effect"[25–27]. Evidence of this effect is clearly displayed by Figs. 2d, f and 3c. However, Fig. 3b seems to display a decrease in amplitude followed by another decrease when propagating past the off and on-ramp respectively. This phenomenon to the author's knowledge has not been reported by other empirical studies. Furthermore, analyzing time-series data from multiple sensors across large distances is regarded as a more challenging problem than a single or local group of sensor data[37]. The works of ref. [37] confirm that similar frequencies are usually detected across nearby sensors and distant sensors usually detect differing frequencies. It is believed that the difference in frequencies across distant sensors is due to the effect that on and off-ramps have on the volume of cars that flow by a specific group of detectors[4]. We emphasize that the Koopman modes we obtain disprove this notion by uncovering patterns that are defined across all detector locations, yet oscillate with a single frequency regardless of the presence of on and off-ramps.

The works of Kim[59] postulate that high-frequency oscillations are more likely to decay in time. Evidence for such a phenomenon can be found by plotting the eigenvalues over the unit circle in the complex plane (Fig. 4a). Eigenvalues whose magnitudes are within a very narrow threshold (0.001) of the unit circle are labeled neutral and correspond to persistent sub-patterns that

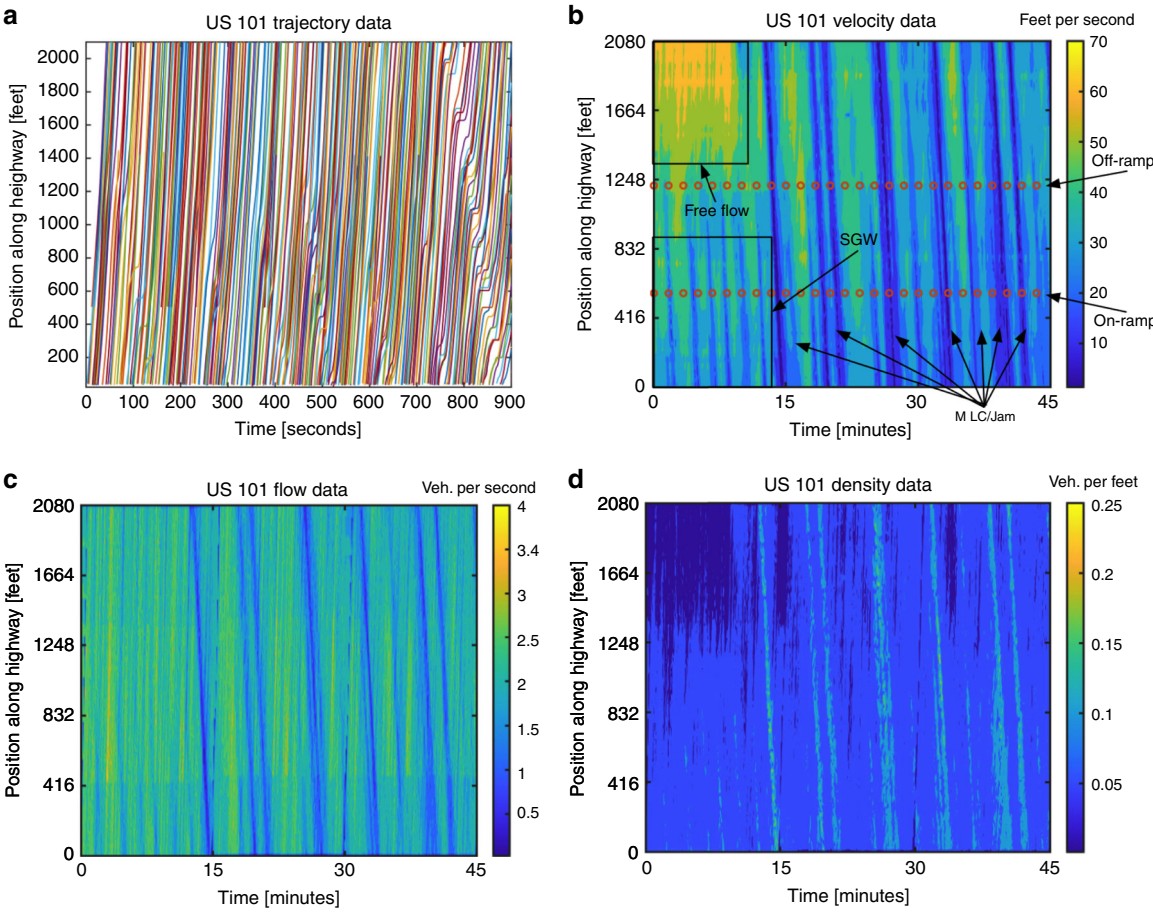

**Fig. 1 Spatiotemporal data for the US-101 highway obtained from binning the NGSIM trajectory data. a** Plot of the trajectory data for the first 15 min. The data was collected between the hours of 7:50 am and 8:35 am, during the onset of congestion. The section of the highway studied consists of five main lanes, a single on and off-ramp and an auxiliary lane between the on and off-ramp. Every colored line corresponds to a unique vehicle. **b** Spatiotemporal velocity data. The locations of the ramps have been labeled with dark-orange dotted lines. During the first 12 min, the post-off-ramp section of the highway experiences a period of free-flowing traffic. However, during this same time, the mid and pre-on-ramp sections of the highway are experiencing stop and go wave traffic, labeled as SGW. During the last 30 min of the study, the highway experiences a series of moving localized clusters labeled MLC, which correspond to traffic jams. **c** Spatiotemporal flow data. The flow data is obtained as the product of the velocity and density data sets. **d** Spatiotemporal density data. As expected, the density appears to be related to the inverse of the velocity profile. Specifically, one can observe that periods corresponding to free-flowing traffic have smaller density and periods corresponding to traffic jams are a result of high density. The source data underlying (**b–d**) are provided in the Source Data file.

neither decay nor grow in time. Likewise, eigenvalues with magnitude larger (or smaller) than this threshold fall outside (or inside) the unit circle and have been labeled as unstable (or stable). What can be seen from Fig. 4a is a cluster of neutral eigenvalues on the far right side (within the dark-gray box) of the unit circle corresponding to the slowest frequencies. This confirms the works of ref. [59] in that the slowest evolving patterns persist in time. Furthermore, the works of Gartner[60] find that patterns associated with longer periods of oscillation are typically accompanied by larger amplitudes. We find evidence for this trend by plotting the average amplitude of each Koopman mode against its period (Fig. 4b). Indeed, one can clearly see a drop in amplitude for decreasing periods of oscillation.

In addition to the daily and weekly cycles, the works of Dendrinos[61] demonstrate the existence of intra-day (less than 24 h) patterns[61]. Plotting the periods of oscillation for the first 15 modes (Fig. 4c) clearly verifies this phenomenon. Further evidence of the existence of intra-day as well as intra-week patterns can be referenced in Supplementary Figs. 13–15. Furthermore, the periods of oscillation we identify are stable

across various choices of observation such as velocity, density, flow, or a concatenation of them all. Lastly, we demonstrate that the modes we recover are indeed physically relevant to the dynamics by plotting the modes corresponding to the stable, unstable and neutral eigenvalues separately (Fig. 4d–f) and then superimposing them (Fig. 4g–h) along with the previously removed average (Fig. 4i) to reconstruct the original data. This demonstrates that the modes we have uncovered are dynamically important sub-patterns and when superimposed together reconstruct the data.

**Koopman mode analysis of multi-lane traffic data.** We demonstrate how the KMD readily generalizes to the multi-lane scenario by analyzing multi-lane spatiotemporal density data for the US-101 highway. The multi-lane data was generated by binning the individual lanes of the NGSIM data. With the addition of an extra spatial coordinate (Lane #), the Koopman modes are now two-dimensional spatiotemporal patterns and best visualized as a video which can be referenced in Supplementary Videos 1–14.

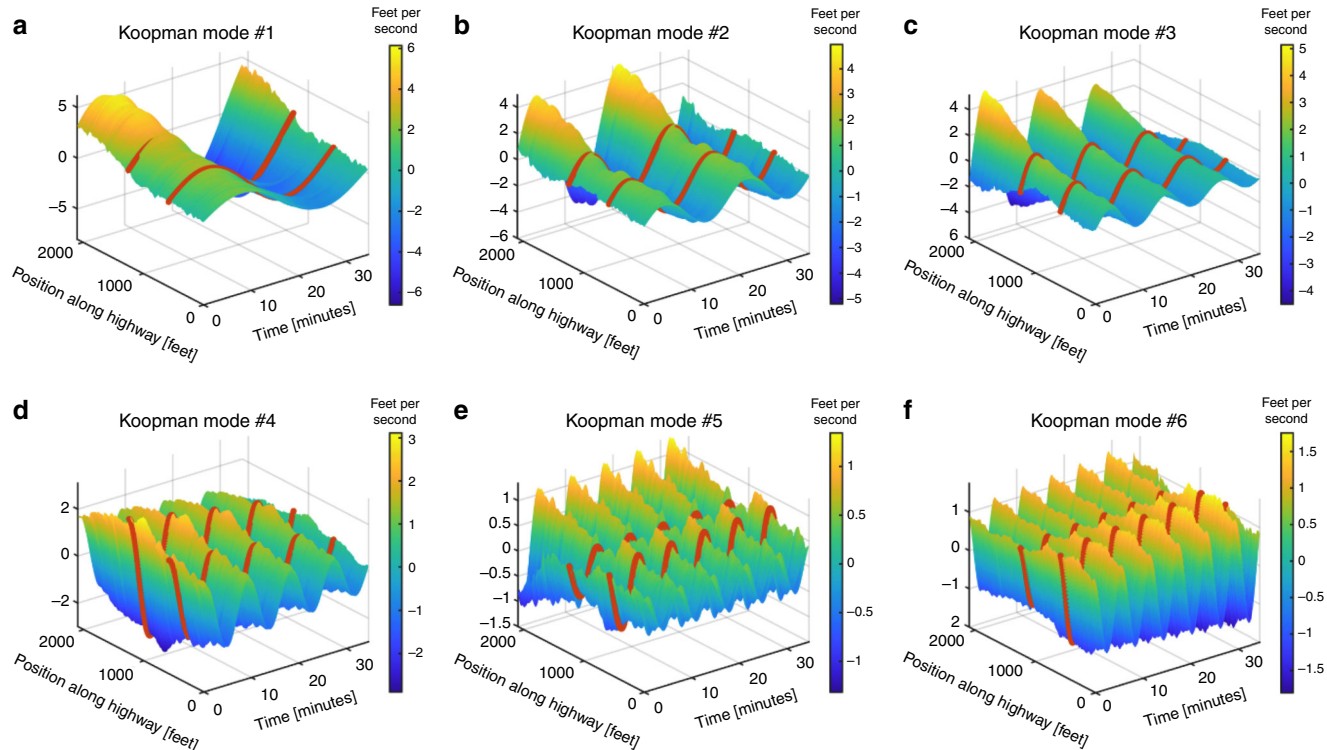

**Fig. 2 Koopman modes demonstrating our method's ability to uncover patterns hidden within traffic velocity data.** The on/off-ramp locations have been labeled with dark orange dotted lines. Modes 1–3 have a very localized structure near the post-off-ramp section, indicating that they correspond to pinned localized clusters (PLC). Furthermore, it is evident that the first three modes capture the general transition from high to low velocities that occurs during the onset of traffic. Mode 5 seems to be a spatial harmonic of the first three modes in that it has another peaked structure in the mid-ramp section of the highway. Modes 4 and 6 provide clear evidence for the pumping effect, where an apparent increase in amplitude followed by a decrease can be seen in these modes as they propagate past the off and on-ramps respectively. Overall, the Koopman modes uncover complex spatiotemporal wave structures that are hidden within traffic data. Furthermore, every mode oscillates with a single known frequency, according to the imaginary part of its corresponding eigenvalue. This can be contrasted to a Fourier analysis which would yield modes and frequencies specific to the positions along the highway. A complete list containing the periods of oscillation of the modes we discussed can be referenced in Supplementary Table 1.

However, we have also provided figures containing snapshots of the videos over an entire cycle in Supplementary Figs. 16–22. The first multi-lane mode (Supplementary Fig. 16) is a spatially localized PLC about the post-off-ramp section of the highway similar to the first mode of the corridor-wide (single lane) analysis. The second and the third mode (Supplementary Figs. 17 and 18) are again harmonics of the first. Interestingly, modes 4, 5, and 10 (Supplementary Figs. 19–21) display a dynamic lane-changing (zig-zag) motion within the mid-ramp section where lane-changing maneuvers are highest due to merging/diverging vehicles. The seventh mode is plotted in Fig. 5. From top-left (Fig. 5a) to bottom-right (Fig. 5f) one can clearly observe how the seventh mode corresponds to an MLC that affects the entire highway. However, the MLC's travel is out of phase across the different lanes, giving the jam an apparent top-left to the bottom-right direction of travel. We encourage the reader to reference Supplementary Videos 1–14, available online, to properly visualize the results of the multi-lane modes.

Lastly, one can see how the merging on-ramp densities (multiplied by 5 for visual purposes) and highway densities are out of phase. This indicates a successful timing of the ramp metering by verifying that incoming vehicles are not allowed to merge while the highway is jammed. The proper implementation of ramp metering algorithms has been shown to significantly improve traffic congestion, reduce travel times, and reduce accidents between merging and flowing traffic[62–64]. However, the proper tuning of the control algorithm's parameters and identification of congestion patterns is critical for a successful implementation. Figure 5 and Supplementary Figs. 6–12 demonstrate how our analysis can identify multi-lane and on-ramp congestion patterns along with their associated time-scales, indicating how a multi-lane KMD analysis can be utilized to verify the proper timing of static ramp meters and incorporated into the development of dynamic ramp metering algorithms.

**Forecasting the California performance measurement system.** The California Department of Transportation (Caltrans) Performance Measurement System (PeMS) data set is a real-time monitoring system for hundreds of highways across the state of California. This measurement system processes 2 GB of real-time data per day and provides access to years of historical data. The historic and real-time nature of the PeMS repository has led to its widespread use for implementing and verifying forecasting methodologies[5,36]. We have developed a moving horizon Hankel dynamic mode decomposition (MH-HDMD) algorithm which utilizes a subset of $s$ data vectors (sampling window) to forecast the next $f$ data vectors (forecast window) updated every $h$ time steps (horizon window). The specific highways we study are as follows. First, we forecast a week and month's worth of data for 100 and 300-mile sections of the eastbound Interstate 10 highway (I-10) and northbound Interstate 5 highway (I-5), respectively. Next, a network of the 10 largest highways connecting Los

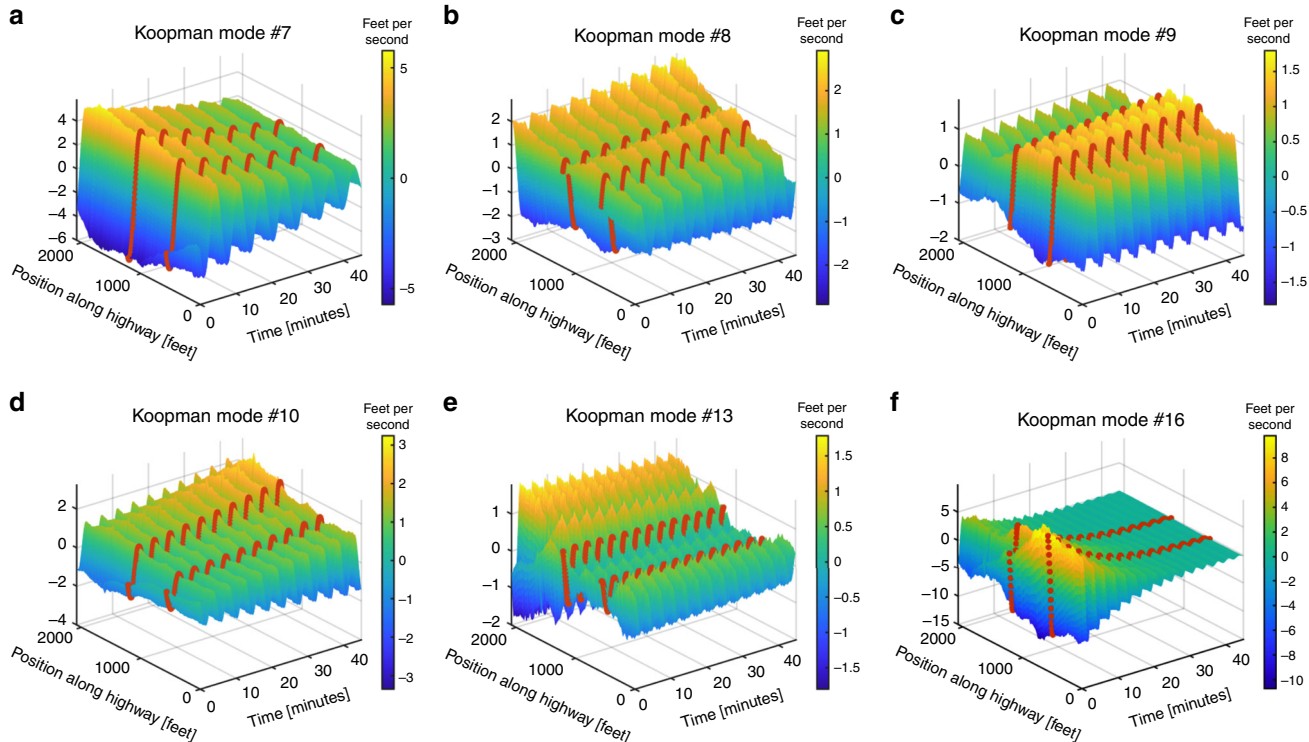

**Fig. 3 Koopman modes demonstrating the ability to uncover growing or decaying patterns.** The on/off-ramp locations have been labeled with dark orange dotted lines. Mode 7 propagates across the entire highway without disturbances to its amplitude and thus corresponds to a highway wide traffic jam also known as a moving localized cluster (MLC). Mode 9 provides further evidence for the pumping effect, where an apparent increase in amplitude followed by a decrease can be seen as the mode propagates past the off and on-ramps respectively. However, mode 8 seems to display a decrease in amplitude followed by another decrease when propagating past the off and on-ramp respectively. This phenomenon to the author's knowledge has not been reported by other empirical studies. Modes 10 and 16 demonstrate our method's ability to uncover growing or decaying patterns. Specifically, mode 16 appears to contain its amplitude almost entirely during the first 10–15 minutes and concentrated in the pre-on-ramp location of the highway. This indicates that mode 16 corresponds to the stop and go waves (SGW) present during the exact same region of the spatiotemporal data in Fig. 1a. The exact growth or decay rate of the mode is dictated by the real part of its corresponding eigenvalue. This again is a distinguishing feature of our methodology from a Fourier analysis in that Fourier modes do not capture growing or decaying features. Lastly, mode 13 also demonstrates a double-peaked structure resembling a spatial harmonic feature of Fig. 2a–c. A complete list containing the periods of oscillation of the modes we discussed can be referenced in Supplementary Table 1.

Angeles to the greater southern California area is forecasted for the week of Christmas 12/21–12/26 of 2016. Additionally, we forecast 100 miles of the northbound US-101 highway during the deadly[65] southern California rainstorm that occurred on February 17, 2017. The data for this example was collected for the days of February 16–17, 2017. In all cases, we utilize the last 15 min to forecast the next 15 min updating our forecasts every 15 min. The original and forecasted data for the weekly and Christmas holiday network data sets are shown in Fig. 6. The original and forecasted data for the I-5 and US-101 data sets can be referenced in Supplementary Figs. 23 and 24.

We quantify the performance of our method by computing the mean absolute error (MAE), mean relative error (MRE), root mean squared error (RMSE), the spatial and temporal averages of the mean absolute error (SMAE and TMAE) and the spatial and temporal correlations (SCorr and TCorr) according to formulas (18)–(22) described within the Methods section. The SMAE and TMAE for the 1-week I-10E, 1-month I-5N, and holiday network data sets are plotted in Fig. 7a–f. It is evident that our method obtains an average MAE between 1 and 2 miles per hour across all detectors, for weeks/months data, across differing highways and network of highways. Furthermore, since the unpredictability of highway traffic renders an absolutely perfect forecast impossible we plot, in Fig. 7g–i, the probability distributions of

the original and forecasted velocities. It is clear to see the near-perfect matches between the statistics of the raw traffic data with our forecasted data. Furthermore, It is evident that we are able to match the distribution of higher velocities with much more accuracy than lower velocities. Nevertheless, the subplots of the lower velocity distributions demonstrate that although states of congestion are inherently more unpredictable[21] our forecasts match the statistics of the data. This indicates that despite the higher variability that is believed to exist within congested traffic there are wave-like patterns that account for a majority of the system dynamics. A similar error analysis for the US-101 highway data set can be referenced in Supplementary Fig. 25 and a complete summary of our error analysis for all highways studied can be referenced in Supplementary Table 1.

We can further investigate how the mean absolute error varies for different choices of $(s, f)$, by producing forecasts for various values of sampling and forecasting windows that are multiples of 15 minutes. For every choice of $(s, f, f)$ we record the MAE and generate a colormap for the I-10 highway (Fig. 8a) and the US-101 (Fig. 8b). It is intuitive that for a fixed sampling rate our error should increase with longer forecast windows, a phenomenon previously observed by others[21]. The counter-intuitive aspect of Fig. 8a–d is that for a fixed forecasting window increasing the size of the sampling window hinders our forecasts. This is best seen by

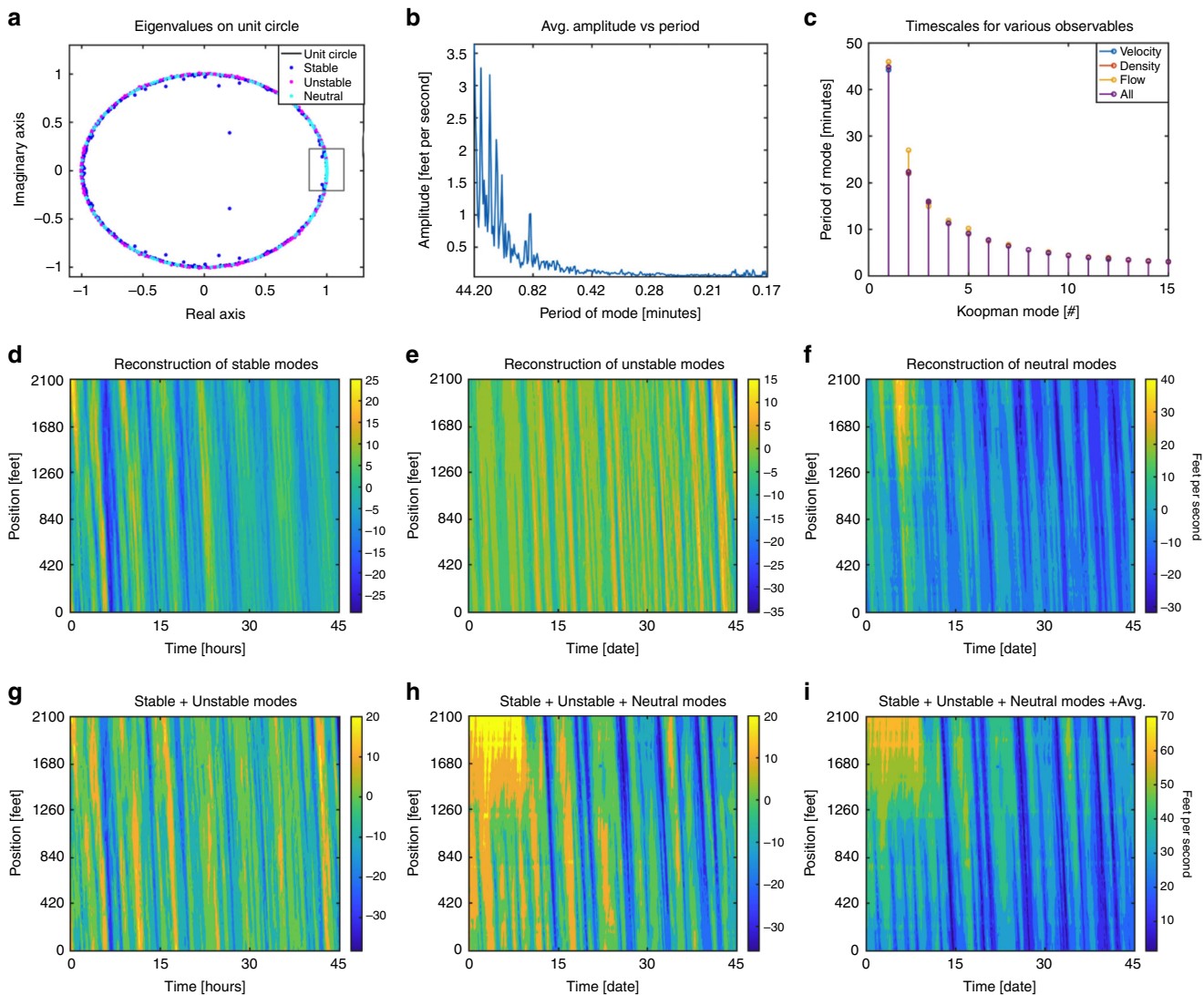

**Fig. 4 Koopman eigenvalue analysis and the corresponding stable, unstable, and neutral modes. a** Plotting the color-coded eigenvalues on the unit circle confirms the findings of Kim[59] in that slowly evolving patterns persist in time. **b** Plotting the average amplitude of the modes against their period confirms the findings of Gartner[60] in that slower evolving patterns carry larger amplitudes. **c** Plotting the timescales identified for various choices of data demonstrates the KMD to be a robust methodology for extracting fundamental frequencies regardless of the modality of observation and verify the existence of intra-day patterns. **d–i** Plots of the corresponding stable, unstable, and neutral modes and various superpositions of them. These figures verify that the individual modes correspond to coherent and dynamically important sub-patterns and when superimposed reconstruct the data.

looking across rows and observing how the error increases. The results in Fig. 8 suggest that the accurate forecasting of traffic is dependent on the most current traffic conditions and not necessarily on the historical past. Indicating that costly training over extensive amounts of historical data is not necessary and in fact, may hinder the ability to forecast. Although this is directly contrary to what many researchers believe[21], it is, in fact, beneficial as it indicates that accurate forecasts can be obtained efficiently with limited data.

**Forecasting and analysis of multi-lane network traffic data.** In this section, we demonstrate how the KMD can be utilized to analyze and forecast highway traffic at the multi-lane network scale. We do so by applying the KMD to highway occupancy data for a network of highways within Los Angeles. The highway occupancy is a normalization of the highway density by the maximum density of the highway. We plot in Fig. 9a a map, taken from Map data ©2019 Google, of Los Angeles with the highways studied highlighted, a plot of the 24-hour Koopman modes

(Fig. 9b) along with the average phase and magnitude (Fig. 9c, d). The red horizontal lines in Fig. 9d serve to visually divide the data corresponding to differing highways. From the 24-hour mode itself, we can immediately observe that the onset of congestion occurs in a certain order within the network. Specifically, the I-105W, I-10W, and I-405N are congested first then the I-710N followed by the I-110N. One can conclude this by observing that the (green) areas of congestion for every highway are staggered, the order of their staggering reveals the order in which they are congested. The order of congestion indicated corresponds with the well-known fact that morning traffic travels in the direction of Los Angeles from San Bernardino County (east to west) and from Orange county (southeast to northwest). This is further verified by observing the near equal phases for the west and northbound directions coming into LA (Fig. 9c) and the near equal, but opposite, phases for the east and southbound directions leaving LA (Fig. 9i). Furthermore, Fig. 9d clearly displays the highest level of magnitude within the I-10E and I-105E highways indicating that, on a 24-h basis, traffic congestion is heaviest along these

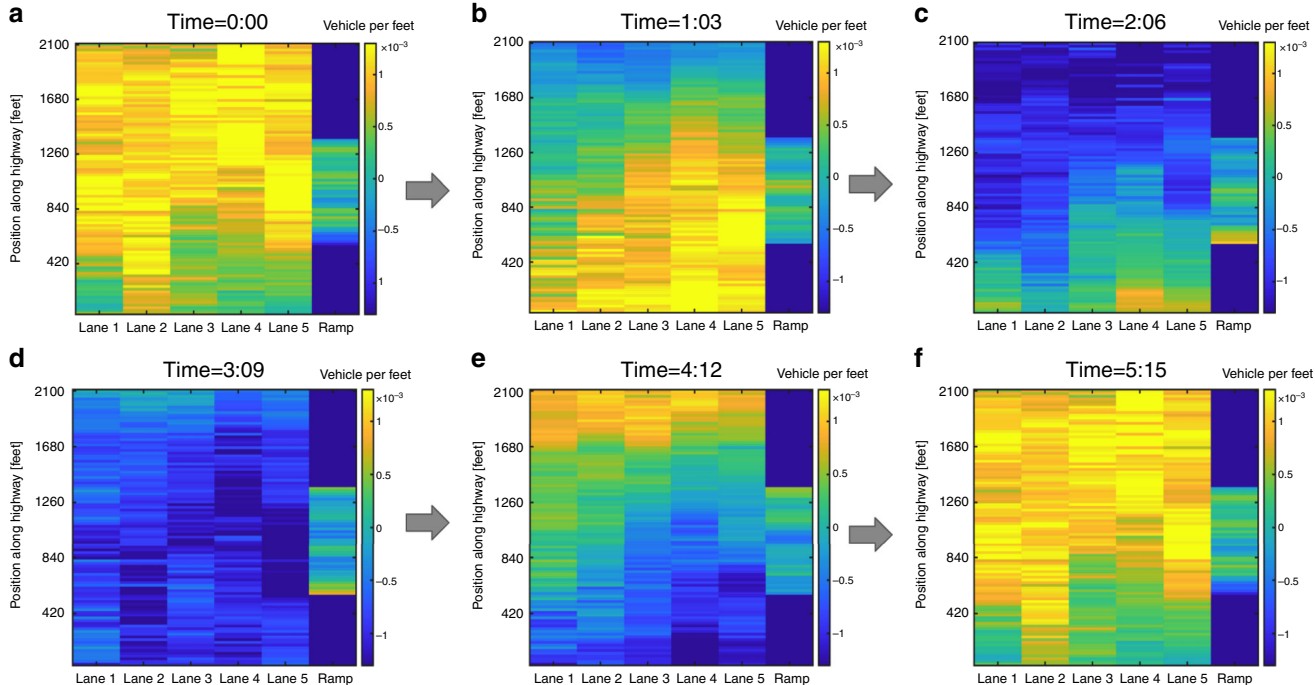

**Fig. 5 Video snapshots of the seventh multi-lane Koopman mode.** The mode has a period of approximately 6 minutes and the time for each figure is given in minutes:seconds. Mode 7 clearly captures the dynamics of a highway wide traffic jam. It is interesting to note how the moving localized cluster's (MLC) travel can be out of phase across differing lanes. This results in the apparent top-left to a bottom-right direction of travel and is a feature impossible to recover from a single-lane analysis. Lastly, one can observe that the on-ramp density of vehicles is not merging at the time of peak congestion. Specifically, the on-ramp is most heavily congested during (**c** and **d**) at which point the MLC has already propagated by. However, mode 14, a harmonic of this mode, clearly displays the opposite effect. This demonstrates how our multi-lane analysis can be utilized to verify the successful timing of static ramp metering algorithms and identify the correct timescales for dynamic ramp metering algorithms. Mode 14 can be referenced in Supplementary Fig. 22.

highways than all others within the network. This is also in line with the fact that the I-10E and I-105E highways connect LA to Orange and San Bernardino County. Interestingly, it is the eastbound direction corresponding to the afternoon rushes that are most heavily congested. This suggests that there are more vehicles traveling in the outbound direction of LA in the afternoon than the original amount of morning commuters that came into LA.

Lastly, we forecast the highway occupancy data for the above-mentioned multi-lane network of highways within Los Angeles. Again, we utilize the last 15 min to forecast the next 15 min and visualize our results as a video, which can be referenced in the Supplementary Video 15. We plot snapshots from the multi-lane network forecast video for the morning (Fig. 10a, b) and afternoon (Fig. 10c, d) rush hours. The forecasted and true traffic conditions at 5:45 am and 6:00 pm are shown. For every plot within Fig. 10, the top and bottom horizontal highways correspond to the (I-10E, I-10W), and (I-105E, I-105W), respectively. The left, center, and right vertical highways correspond to the (I-405S, I-405N), (US-110S, US-110N), and the (I-710S, I-710N), all of these highways are highlighted within Fig. 9a. Both morning and afternoon forecasts demonstrate a high level of similarity with the true conditions. However, the forecasts were available between 5 and 15 min prior to the actual conditions occurring. The corresponding error analysis, shown in Fig. 10e–g, validates that our forecasts remain accurate over the entire day. We encourage the reader to view the entire video of the forecasting results for the Los Angeles multi-lane network which is available online in Supplementary Video 15.

## Discussion
We have proposed a data-driven method, based on the spectral properties of the Koopman operator, which provides a

platform for the identification and analysis of spatiotemporal traffic patterns. We were able to distinguish between the various types of patterns previously proposed by Ahn, Laval, and others[20,26,27] (MLC, PLC, SGW, "pumping effect"). We identify new patterns with standing wave node-like features, spatial harmonic features, growing/decaying patterns, multi-lane MLC patterns, multi-lane PLC patterns, multi-lane patterns with combined lateral and longitudinal (zig-zag) travel, multi-lane patterns associated with the merging effects of on-ramps and novel patterns that exist within a network of highways. We show that network mode analysis can reveal the order of congestion, synchrony of congestion, and which highways are occupied the most. Every pattern we uncovered is global (across all detectors) and oscillates with a single corresponding frequency, in contrast to Fourier transform methods that would detect different frequencies for different sensors. Furthermore, via Koopman eigenvalue analysis, we provide objective means for extracting temporal characteristics of traffic patterns and further analyze the eigenvalues to provide evidence for the works of Kim, Gartner, Dendrinos[30,37,59–61], and others. In addition to the well-known daily and weekly cycles, we have demonstrated the existence of intra-week (Supplementary Figs. 14 and 15) and intra-day patterns within traffic. Lastly, we have demonstrated how superimposing the modes accordingly can yield a decomposition of the data into the corresponding decaying, growing and persisting sub-patterns.

We have also developed an accurate and efficient platform for the real-time forecasting of highway traffic conditions. Our method demonstrates that the wave-like trends account for a majority of the dynamics and yield accurate forecasts, despite the unpredictability involved in a traffic system. As opposed to

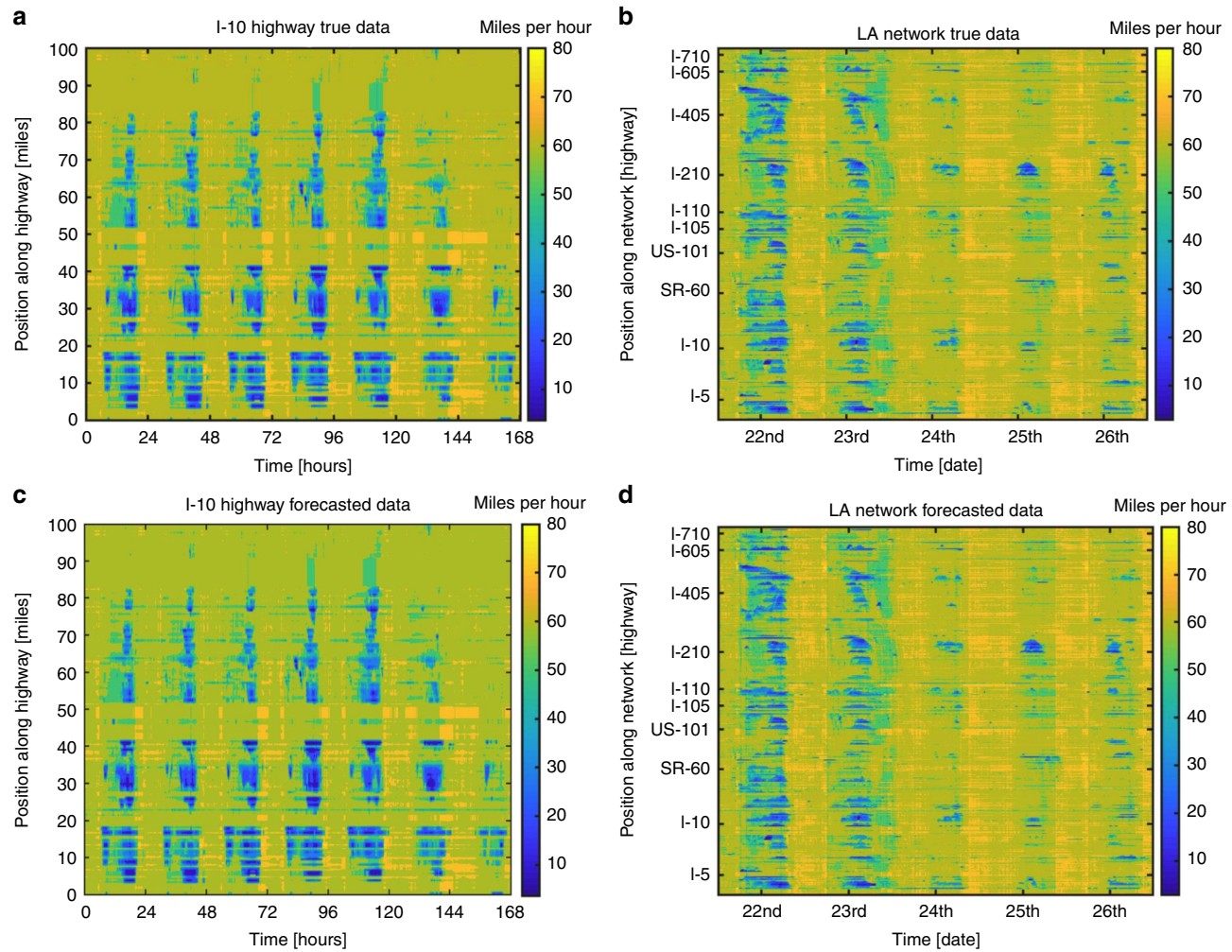

**Fig. 6 Comparison of the true and forecasted data for the I-5 and LA network. a**, **b** Raw data collected from PeMs for the I-10E and southern California network. The data was collected for a week and 5 days respectively. The I-10 highway seems to be mostly congested throughout the week until Sunday. As expected, all highways within the network seem to be congested on the days leading up to Christmas eve (12/21–12/22). Interestingly, there appears to be drastic relief in congestion during and after the actual holiday dates of 12/24–12/26. **c**, **d** Forecasted data generated by the MH-HDMD utilizing the last 15 min of data to forecast the next 15 min. The network was forecasted by concatenating data obtained for individual highways and applying the MH-HDMD algorithm to the concatenated data. By visual inspection alone the raw and forecasted data sets are indistinguishable, indicating an accurate forecast. Nevertheless, there is some error present within our forecasts which can be referenced in Fig. 7. The source data underlying Fig. 7a, b are provided in the Source Data file.

previous approaches we have not filtered, smoothed or aggregated our data, we have not distinguished between weekday/weekend or adverse weather conditioned traffic, nor have we limited our analysis to single or few detectors. Our method's performance does not rely on large historical training nor parameter tuning or selection. We showed the capability of the method to generalize to the challenging scenario of multi-lane highways and multi-lane networks of highways, without any loss to its performance or efficiency. The robustness, efficiency, and versatility of the algorithm make it capable of implementation with real-time monitoring systems to provide cost-efficient forecasts. This is in strong contrast to many state-of-the-art benchmarks that depend heavily on the proper pre-processing of data, tuning of parameters and training over large historical data only to produce case-specific (weekday/weekend/holiday), location-specific and limited (single-lane, single highway) forecasts. The information uncovered by the KMD can be relayed to autonomous vehicle control units as well as dynamic on-ramp metering algorithms to mitigate traffic. Future work will utilize the KMD to develop such

dynamic traffic control algorithms as well as extending the current results to quarterly, yearly, decennial time scales and to urban traffic networks.

Lastly, we emphasize that our methodology makes no assumptions on the physical nature of the underlying system. We only assume to have time-ordered data arising from observations of a linear or nonlinear dynamical system. The forecasting methodology we have developed is in fact quite general and can be applied across different fields of study beyond highway traffic.

## Methods

**Spatiotemporal binning method**. We describe in more detail the binning method utilized in converting the NGSIM vehicle trajectory data. The motivation behind this is that we are interested in identifying spatiotemporal patterns and must construct macroscopic data from the NGSIM trajectory data. To do this we implement the same binning methods developed by ref. [56] and divide the spatio-temporal domain $[0, L] \times [0, T]$ into individual bins of size $\Delta X \times \Delta T$. $L$ represents the total length of the highway, $T$ the total time the data was collected for, $\Delta X$ is the spatial step size, and $\Delta T$ is the temporal step size. Therefore, a single bin is given by the following formula shown in Eq. (1), where $n_x = \frac{L}{\Delta X}$ and $n_t = \frac{T}{\Delta t}$ are the number

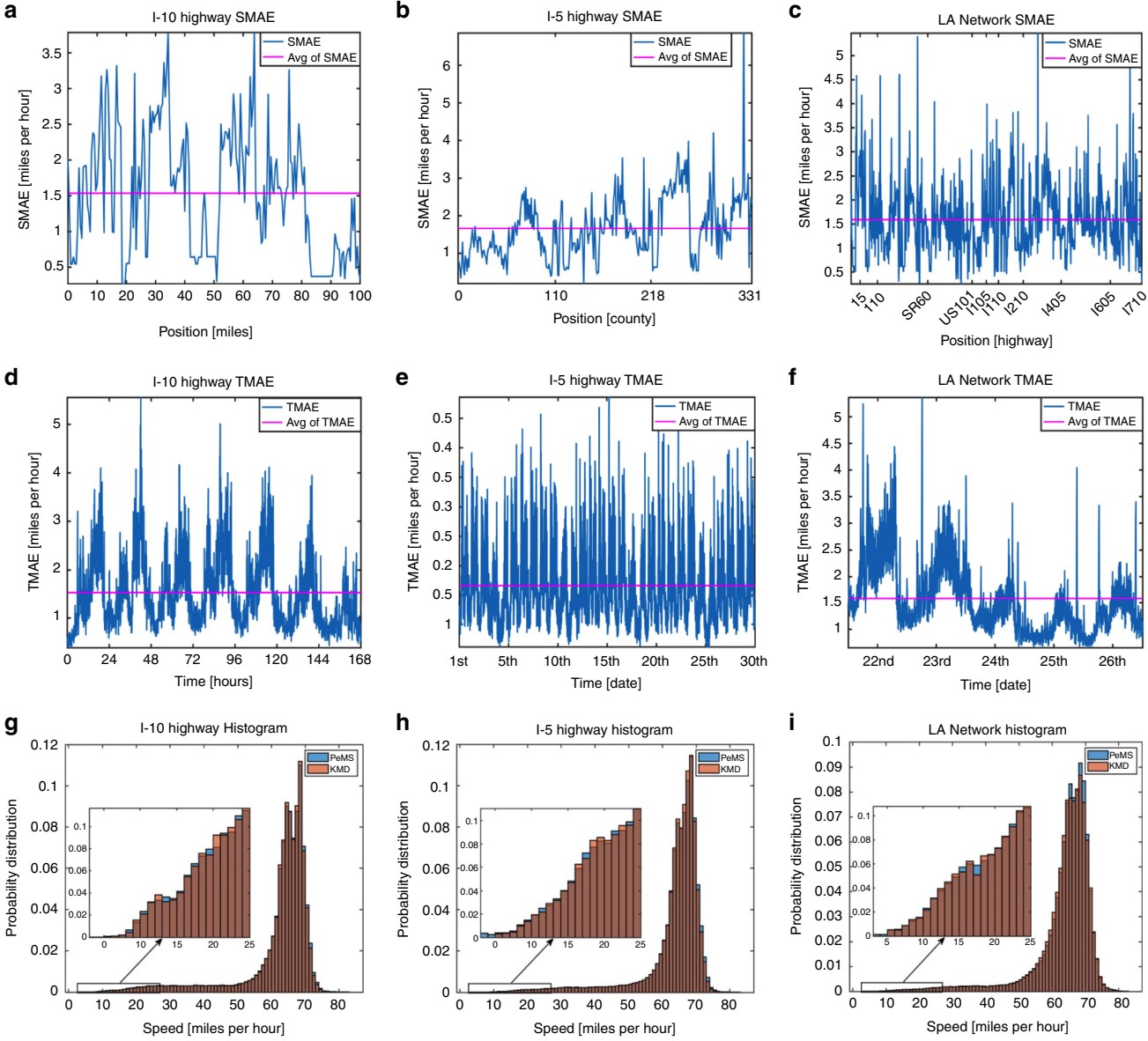

**Fig. 7 Forecasting error analysis indicating that accurate forecasts are generated by the MH-HDMD algorithm. a–c** Spatial MAE across detector locations and its average across all detectors for the I-10, I-5, and Los Angeles network. The SMAE is evidently stable across different highways and networks of highways and is on average between 1 and 2 miles per hour. **d–f** Temporal MAE and its time average. Again, it is evident that our method is not only stable across differing highways and spatial scales but also stable across a wide range of time scales (days, week, month). **g–i** Normalized histograms of highway velocities for both true and forecasted data sets. The near-perfect matches for low and high speeds indicate that despite the unpredictability present in a highway system the statistics of our forecasts are nearly identical with the real physical system. A complete summary of the error analysis for all highways can be referenced in Supplementary Table 2.

of bins in space and time.

$$\text{Bin}_{i,j} = [i\Delta X, (i+1)\Delta X] \times [j\Delta T, (j+1)\Delta T]_{i\in(0\dots n_x), j\in(0\dots n_t)} \quad (1)$$

For every bin the quantities of interest (velocity, density, or flow) are assumed constant. This assumption allows one to use the number of traces (footprints) of vehicles left within a bin to estimate the macroscopic speed, density, and flow. Specifically, the speed is computed as the average of all velocity traces left within a bin according to Eq. (2), where $\text{trace}_{i,j} = \{\text{trace} | \text{trace} \in \text{Bin}_{i,j}\}$ which represents the vehicles traces left within $\text{Bin}_{i,j}$ and $\mathbf{V}_{\text{trace}_{i,j}}$ represents the velocity of those traces.

$$\hat{V}_{i,j} = \text{Mean}\left(\mathbf{V}_{\text{trace}_{i,j}}\right) \quad (2)$$

The density is directly related to the amount of time vehicles spend within a given bin normalized by the size of the bin[56–58]. Therefore, the density is computed

according to Eq. (3).

$$\hat{\rho}_{i,j} = \frac{\text{Card}(\text{trace}_{i,j})\text{Sampling Rate}}{\Delta X \Delta T} \quad (3)$$

where Card is the cardinality of a set. Our estimate for the flow is computed according to Eq. (4) and has been shown in ref. [58] to be a reasonable estimate for the flow.

$$\hat{Q}_{i,j} = \hat{V}_{i,j}\hat{\rho}_{i,j} \quad (4)$$

As mentioned, the data collected for the US-101 highway was taken for a total of 45 min between 7:50 am and 8:35 am, however, the data for the I80 highway was collected for a 15-min interval between 4:00 pm and 4:15 pm and again for 30 min between 5:00 pm and 5:30 pm. This leads to the construction of three separate data sets corresponding to the US-101 highway, 4 pm I-80 highway, and 5 pm I-80 highway. Supplementary Figures 1 and 2 demonstrate the resulting velocity, density and flow data sets for both highways for $\Delta X = 20$ feet per bin and $\Delta T = 5$ s per bin.

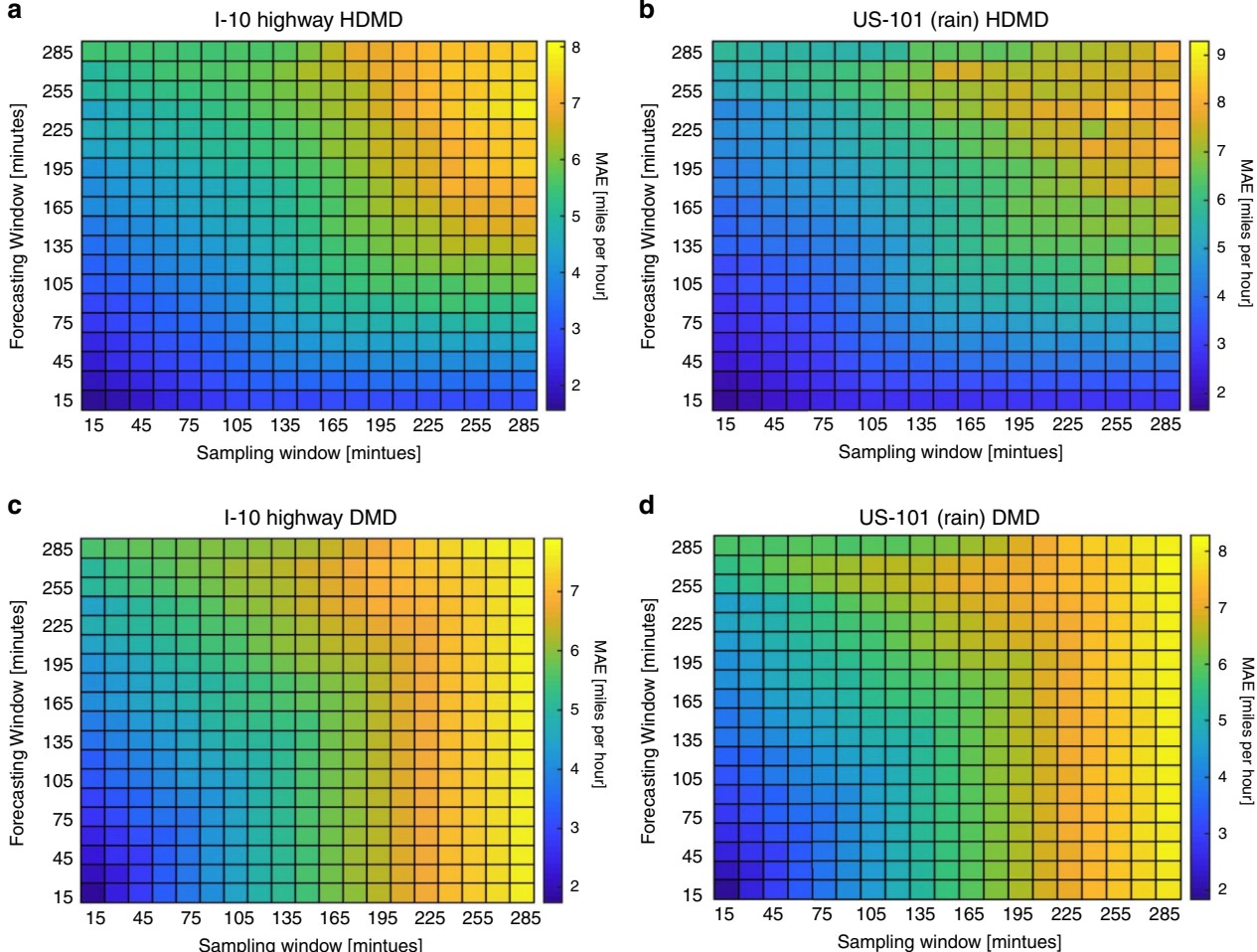

**Fig. 8 Mean absolute error for various choices of (s, f). a, b** The top row was generated utilizing the MH-DMD algorithm. **c, d** The bottom row was generated utilizing the DMD algorithm. It is evident that the Hankel-DMD can provide greater accuracy than a standard DMD analysis. This is best seen by observing the increase of blue-colored squares in (**a, b**) versus (**c, d**). In both cases, the results suggest that the forecasting of traffic is a temporally local task and that current and future conditions depend very little on the historical past. Furthermore, increasing the amount of historical data utilized can hinder the accuracy of forecasts. This demonstrates the need for intelligent transportation systems to have sufficient analytic capabilities of detecting dynamic traffic patterns in real-time and that training over extensive amounts of historical data is not necessary.

Lastly, we alert the reader that the NGSIM data is provided in 15-min intervals. For example, the 45 min of the US-101 data was provided in three separate 15-min interval data sets. Unfortunately, this leads to missing data within the NGSIM data sets. This required us to bin the individual data sets, concatenate the resulting spatiotemporal data and interpolate any missing information. The source code for the binning method and the interpolation has been made available as well as our resulting spatiotemporal data sets. Therefore whether the reader wishes to download the raw trajectory data or our binned spatiotemporal data our results can be reproduced in either situation. A flow scheme demonstrating the binning method outlined in this section can be referenced in Supplementary Fig. 26.

**Koopman mode decomposition of highway traffic dynamics**. In this section, we outline the general theory and setting of the Koopman mode decomposition and its data-driven implementation to highway traffic data. The value versus time point of view of dynamical systems requires one to identify and solve the appropriate differential equation that models the system of interest. The result is a solution of the dynamical system's behavior as a function of time. Although this procedure yields a precise description of the system completely in terms of the initial conditions, it is well known that many differential equations do not admit closed-form analytic solutions. The state-space, point of view of dynamical systems reformulates the underlying differential equations in terms of a vector field defined over an appropriate state space. It happens to be that properties of the vector field and geometrical objects of the state space can reveal an enormous amount of information of the dynamics without needing to solve the equations. While this point of view has been very fruitful for over a century, oftentimes one is interested in studying very high dimensional complex and nonlinear systems where the

equations of motion are not known to begin with. In these situations, one only has access to data or observations of the dynamical system rather than knowledge of the equations that govern the process. The spectral operator point of view provides a means for extracting information of the dynamics strictly from observations or data. This point of view was pioneered by Bernard Koopman whose name has been attributed to the specific operator describing the evolution of observables. The Koopman operator is linear and its spectrum allows one to study nonlinear, complex and high-dimensional systems via linear, although, infinite-dimensional techniques. The epitome of this situation can be taken as highway traffic systems, in which decades of research has been devoted towards deriving the correct governing equations to no avail. The difficulty in deriving such equations and the advent of technology has dramatically shifted the attention of researchers towards data-driven techniques. This has lead to the deployment of a plethora of statistical and machine learning methodologies to analyze and forecast traffic patterns. Unfortunately, many of these methodologies pay no attention to any underlying dynamics but rather seek to simply fit or learn traffic from data. On the other hand, the KMD seeks to uncover the dynamical features hidden within traffic data via the spectrum of a linear operator.

We now explain in detail the mathematical setting of the Koopman operator by considering a continuous-time dynamical system whose state space is an $N$-dimensional smooth manifold $\mathbb{M}$, subject to the dynamics of some vector field $\mathbf{F}$. We could consider a general state-space, which is not necessarily a manifold, however, for ease of exposition we consider the previously mentioned situation. The dynamical system $\mathbf{F}$, is assumed to generate a continuous group of invertible flow maps $\mathbf{S}_t : \mathbb{M} \mapsto \mathbb{M}$, $t \in \mathbb{R}$ which represent the solution of the system. Therefore, for any initial condition $\mathbf{a}_0 \in \mathbb{M}$ its evolution under the dynamics at some later time $t$ is given by $\mathbf{a}_t = \mathbf{S}_t(\mathbf{a}_0)$. Let $\mathcal{H}$ be a Hilbert space of functions and

**a**

Map of network

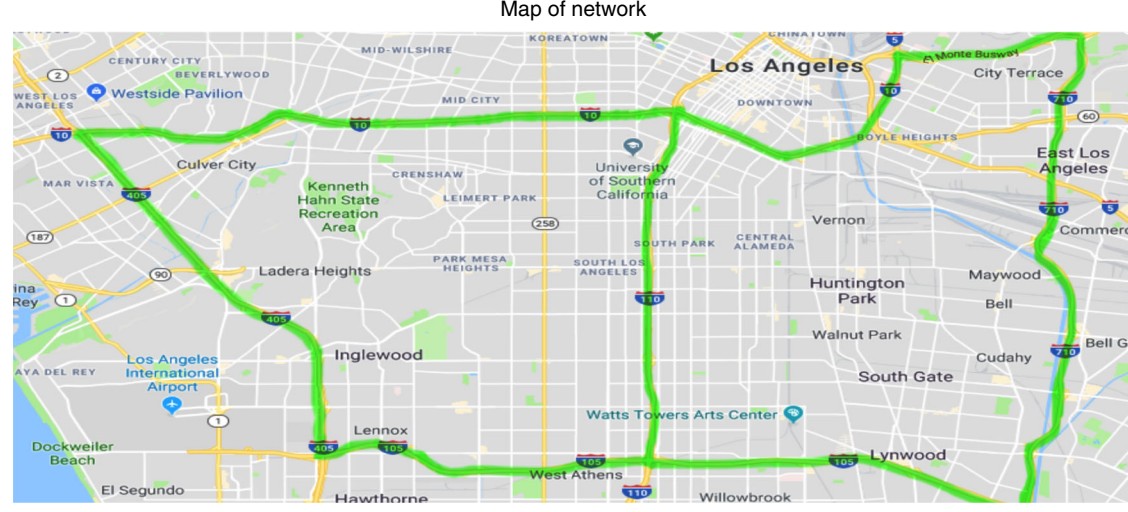

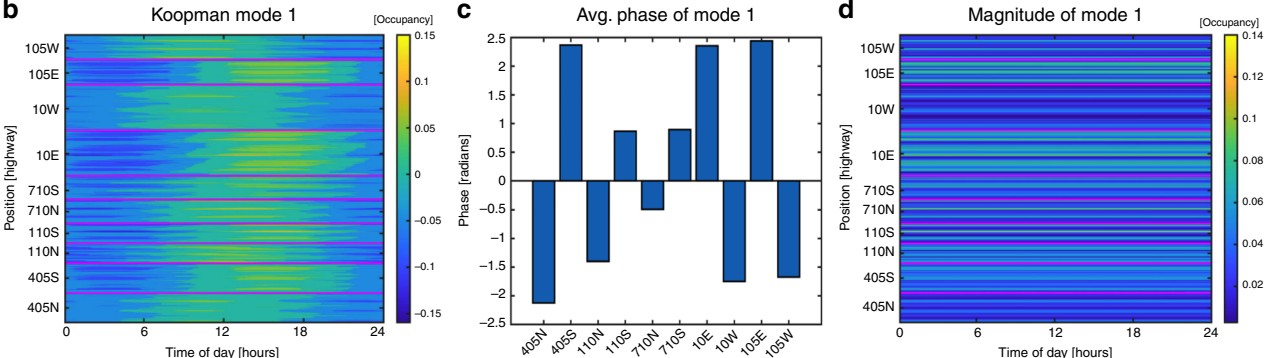

**Fig. 9 Map of the Los Angeles multi-lane network and the resulting 24-hour Koopman. a** Map of the multi-lane network obtained from Map © data 2019 Google. The highways studied are highlighted and data for three lanes of both north–south and east–west directions are collected for the entire day of December 20th, 2018. **b** The 24-hour Koopman mode revealing the order of congestion within the network. Specifically, by observing the staggering of the amplitude one can see that during the morning rush the westbound I-105 and I-10 along with the northbound I-405 and I-110 jam first. In the afternoon traffic switches directions and it is the eastbound and southbound directions of the previously mentioned highways which are jammed. This corresponds with the well-known fact the morning commuters generally travel from San Bernardino (east to west) and Orange County (southeast to northwest) into Los Angeles. **c** A plot of the average phase of the mode sorted by highway confirming the previously mentioned synchrony of congestion. Specifically, the north and west directions seem to be in phase with each other and likewise for the south and east directions. **d** The magnitude of the 24-hour mode along with magenta-colored lines used to visually divide the differing highways. The magnitude of the mode reveals that the eastbound I-10 and I-105 highways are more occupied than the other highways.

let $U^t$ be the Koopman semigroup of operators, parameterized by time $t$, which compose the functions in $\mathcal{H}$ with the flow $\mathbf{S}_t$. Hence, for any function $f \in \mathcal{H}$ and time $t$ the action of the time-$t$ Koopman operator on $f$ can be expressed as follows refs. [39,40]

$$U^t f(\mathbf{a}_0) = f \circ \mathbf{S}_t(\mathbf{a}_0) = f(\mathbf{a}_t). \tag{5}$$

Note that since the Koopman operator is infinite-dimensional, its spectrum may contain a continuous spectrum in addition to the traditional point spectrum of finite-dimensional operators. Assuming that the Koopman operator of a given dynamical system contains only a point spectrum we have that for any $\phi \in \mathcal{H}$ that is an eigenfunction of $U^t$, at eigenvalue $e^{\lambda t}$ for $\lambda \in \mathbb{C}$, its evolution in time is as follows refs. [41–43]

$$U^t \phi(\mathbf{a}_0) = \phi \circ S_t(\mathbf{a}_0) = e^{\lambda t} \phi(\mathbf{a}_0). \tag{6}$$

Furthermore, a subspace $\mathcal{A} \subset \mathcal{H}$ is invariant to the dynamics if for any $f \in \mathcal{A}$, its image under the flow, $U^t f \in \mathcal{A}$ for all time $t$. If the Koopman eigenfunctions or a subset of them, form a basis for an invariant subspace of $\mathcal{H}$, then we can represent any function that lies in the invariant subspace with the eigenfunction basis[41–43]. Formally speaking, let $\Phi = \{\phi_i\}$, $i \in \mathbb{N}$ be a set of Koopman eigenfunctions, $\Lambda = \{\lambda_i\}$ the set of corresponding eigenvalues and let $\mathcal{E}_\Phi = span(\Phi)$ be the Koopman eigenspace associated with the basis of eigenfunctions. Then the Koopman mode decomposition of any observable $f \in \mathcal{A}$ is given by the following

expression[41–43]

$$U^t f(\mathbf{a}_0) = f \circ \mathbf{S}_t(\mathbf{a}_0) = \sum_{i=1}^{\infty} \phi_i(\mathbf{a}_0) e^{\lambda_i t} v_i, \tag{7}$$

where $v_i = \langle f, \phi_i^* \rangle$ is the skew-projection of $f$ onto $\mathcal{E}_\Phi$, obtained via the inner product of $f$ and the dual eigenfunction $\phi^*$, and is called a Koopman mode[41–43]. Although, together the triplet $(\Phi, \Lambda, V)$ yield the KMD of an observable, it is important to note that $(\Phi, \Lambda)$ are intrinsic to the dynamical system however, the Koopman modes $V = \{v_i\}$ are not. Namely, they depend on the choice of $f$ and will change according to that choice.

We now discuss the details of how the Koopman mode decomposition can be applied to highway traffic data. Formally, we have a time-ordered data matrix $\mathbf{X}$ which contains a total of $m$ data vectors. Typically, the velocity or density profile along the highway, at an instant in time $i \in \{1, ..., m\}$, constitutes a single data vector labeled $\mathbf{x}_i$ and corresponds to the $i$th column of the data matrix $\mathbf{X}$. The number of rows in $\mathbf{X}$, labeled by $k$, is dictated by the number of locations along the highway at which the velocity or density is measured (number of sensors). Hence the data matrix has the following form shown in Eq. (8).

$$\mathbf{X} = \begin{bmatrix} \mathbf{x}_1 & \mathbf{x}_2 & \dots & \mathbf{x}_m \end{bmatrix} \tag{8}$$

Again, in Eq. (8) $m$ is the number of data snapshots acquired, $\mathbf{x}_i \in \mathbb{R}^k$, is a single data vector at time $i$ and $\mathbf{X}$ is a $k \times m$ matrix. As previously mentioned, several algorithms for approximating the eigenfunctions, eigenvalues, and modes of the Koopman operator have been developed. In our work we begin by computing

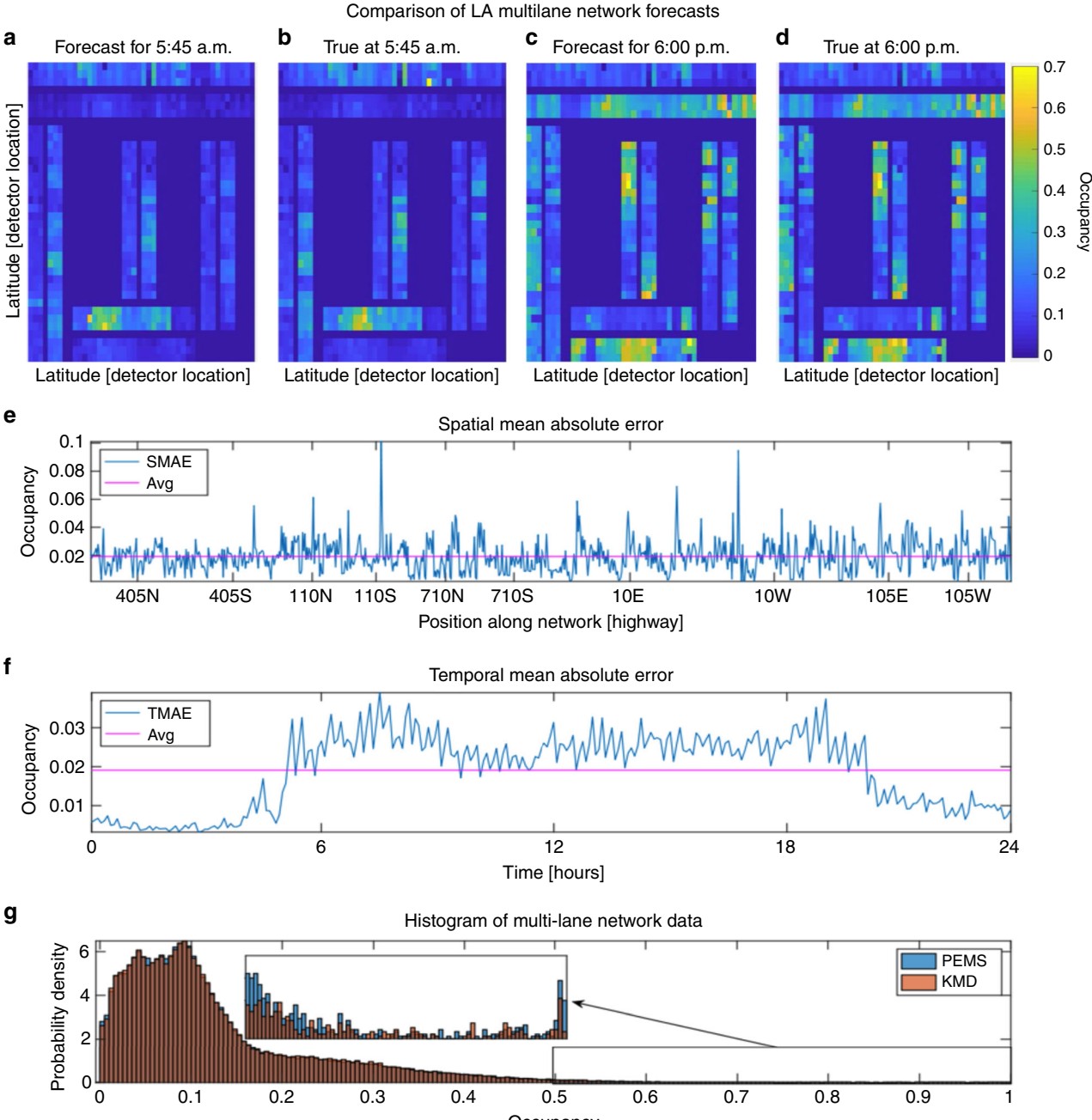

**Fig. 10 Snapshots of the multi-lane network forecast video along with the corresponding error analysis.** For every plot within (**a–d**) the top and bottom horizontal highways correspond to the I-10, and I-105, respectively. The left, center and right vertical highways correspond to the I-405, I-110, and I-710. **e**, **f** Plots of the spatial and temporal mean absolute errors indicating that on average we obtain an error of 2% occupancy, where the highway occupancy is a measure corresponding to the normalization of the highway density by the maximum density of the highway. **g** Histogram of the raw and forecasted data sets indicating that the probability density functions of our forecasts match the statistics of the system. We encourage the reader to reference the complete video of the forecasts available online in Supplementary Video 15.

the mean subtracted data matrix $\hat{\mathbf{X}}$ according to Eq. (9).

$$\hat{\mathbf{X}} = \left[ \mathbf{x}_1 - \frac{1}{m}\sum_{i=1}^{m}\mathbf{x}_i \quad \mathbf{x}_2 - \frac{1}{m}\sum_{i=1}^{m}\mathbf{x}_i \quad \cdots \quad \mathbf{x}_m - \frac{1}{m}\sum_{i=1}^{m}\mathbf{x}_i \right] \quad (9)$$

It is clear to see that $\hat{\mathbf{X}}$ is obtained by computing the time averages of the original data and subtracting the computed average from the data. This is motivated by the fact that $\lambda = 1$ is an eigenvalue of the Koopman operator and corresponds to the time averages of the system[41–43] and hence we pre-compute this quantity. The mean-subtracted data matrix is then fed to the Hankel-DMD algorithm[49,66] which is a combination of a time-delay (Hankel matrix) embedding that is followed by an exact dynamic mode decomposition algorithm[51] (Exact-DMD). The method of delay embedding is a state-space reconstruction technique

that has been shown to recover the attractor (structure) of the original dynamical system generating the data[67,68]. Hence, a delay embedding $d$ is chosen and the mean subtracted data is embedded as shown in Eq. (10).

$$\mathbf{X} = \left[\mathbf{x}_1 \quad \cdots \quad \mathbf{x}_m\right] \rightarrow \mathbf{H} = \begin{bmatrix} \mathbf{x}_1 & \mathbf{x}_2 & \mathbf{x}_3 & \cdots & \mathbf{x}_{m-d} \\ \mathbf{x}_2 & \mathbf{x}_3 & \mathbf{x}_4 & \cdots & \mathbf{x}_{m-d+1} \\ \vdots & \vdots & \vdots & \ddots & \vdots \\ \mathbf{x}_d & \mathbf{x}_{d+1} & \mathbf{x}_{d+2} & \cdots & \mathbf{x}_m \end{bmatrix} = \left[\mathbf{h}_1 \quad \cdots \quad \mathbf{h}_l\right]$$

$$(10)$$

Now, what the Exact-DMD seeks to approximate is a finite-dimensional representation of the Koopman operator which must satisfy the following relation

shown in (11).

$$\mathbf{H}_2 = \mathbf{K}\mathbf{H}_1 + \mathbf{r} \tag{11}$$

where $\mathbf{K}$ is a finite matrix representation of the Koopman operator and $\mathbf{r}$ is a residual error term due to the fact that we only have a finite-dimensional approximation of a possibly infinite expansion. $\mathbf{H}_1$ and $\mathbf{H}_2$ are time-shifted matrices as shown in Eqs. (12) and (13)

$$\mathbf{H}_1 = [\mathbf{h}_1 \quad \mathbf{h}_2 \quad \dots \quad \mathbf{h}_{l-1}] \tag{12}$$

$$\mathbf{H}_2 = [\mathbf{h}_2 \quad \mathbf{h}_3 \quad \dots \quad \mathbf{h}_l] \tag{13}$$

The Exact-DMD obtains this approximation by minimizing the residual term in a least squares sense. Therefore, by utilizing the singular value decomposition (SVD) of $\mathbf{H}_1 = \mathbf{U}\Sigma\mathbf{W}^*$ to rewrite (11) as shown in Eq. (14).

$$\mathbf{H}_2 = \mathbf{K}\mathbf{H}_1 + \mathbf{r} = \mathbf{K}\mathbf{U}\Sigma\mathbf{W}^* + \mathbf{r} \tag{14}$$

Multiplying both sides of (14) with $\mathbf{U}^*$, and recalling that minimizing the residual term requires it be orthogonal to $\mathbf{U}$ we obtain the following expression in Eq. (15).

$$\mathbf{U}^*\mathbf{H}_2 = \mathbf{U}^*\mathbf{K}\mathbf{U}\Sigma\mathbf{W}^* + \mathbf{U}^*\mathbf{r} = \mathbf{U}^*\mathbf{K}\mathbf{U}\Sigma\mathbf{W}^* \tag{15}$$

Rearranging the above equation we can obtain a matrix $\mathbf{S}$ that is related to $\mathbf{K}$ via a similarity transformation as shown in (16).

$$\mathbf{U}^*\mathbf{H}_2\mathbf{W}\Sigma^{-1} = \mathbf{U}^*\mathbf{K}\mathbf{U} \equiv \mathbf{S} \tag{16}$$

Since $\mathbf{K}$ and $\mathbf{S}$ are related they share common eigenvalues and the eigenvectors are the same up to the similarity transformation. Hence if $(\lambda_i, \mathbf{w}_i)$ are an eigen-pair of $\mathbf{S}$ then $(\lambda_i, \mathbf{v}_i = \mathbf{U}\mathbf{w}_i)$ is an eigen-pair of $\mathbf{K}$. Furthermore, since the sampled data produced a discrete time description of an originally continuous time process, the eigenvalues $\{\lambda_i\}$ we obtained lie on the unit circle. Therefore, the continuous time eigenvalues are given by $\omega_i = \frac{ln(\lambda_i)}{T}$, where $T$ is the sampling rate. Finally, we can use the KMD to obtain a description of the observed data points $\mathbf{x}_i$ via the following equation:

$$\mathbf{x}_{\mathrm{kmd}}(t) = \sum_{i=1}^{l} b_{0i}v_i e^{\omega_i t} = \mathbf{V}e^{\omega t}\mathbf{b}_0 \tag{17}$$

where $\mathbf{V}$ is a matrix whose columns are the eigenvectors $v_i$ and $\mathbf{b}_0$ is a vector of coefficients associated with the initial data snapshot $\mathbf{x}_1$, specifically $\mathbf{b}_0 = \mathbf{V}^\dagger\mathbf{x}_1$. Where $\dagger$ represents the Moore–Penrose pseudoinverse of a matrix and $e^{\omega t}$ represents a diagonal matrix whose elements are $e^{\omega_i t}$.

Therefore a pair $(v_i, \omega_i)$ of Koopman modes and eigenvalues are obtained from the Hankel-DMD algorithm. The modes and eigenvalues can then be evolved in time via Eq. (17) for $m$ time steps to reconstruct the data and by iterating past $m$ time steps one can begin to forecast the future state of the system. A flow scheme of the procedure we outlined can be referenced in Supplementary Fig. 27. Lastly, the works of ref. [69] have shown that applying a DMD algorithm to mean subtracted data reduces to a standard discrete Fourier transform of the data. We emphasize that this is not the case in our situation due to the time-delay embedding. In our works, we first subtract the mean and then embed before applying Exact-DMD. One can check the structure of the Hankel matrix in Eq. (10) and confirm that although the mean was subtracted from the rows of the original data the mean of the rows of the Hankel matrix is not zero. This is due to the rearranging of the elements that takes place when embedding and forming the Hankel matrix. Indeed, had we first embedded and then subtracted the mean, in that situation, we would be applying DMD to a mean subtracted data. Essentially, the order in which one mean subtracts and embeds is crucial. For further discussion on the Koopman operator, dynamic mode decomposition, Hankel matrices, and time-delay methods, refer to refs. [46,49,50,55,67,68,70]. A detailed discussion on how we select the number of delays can be referenced in the following section. Pseudocode of the Hankel-DMD algorithm can be referenced in Supplementary Note 1 and the corresponding source code is made available according to the code availability statement.

**Choice of hyperparameters**. We emphasize that the spatiotemporal bin size $\Delta X$ and $\Delta T$ used to convert the Lagrangian NGSIM data into Eulerian data is not relevant to our methodology but specific to how the data was provided. For example, the Caltrans PeMs data was provided as Eulerian data, to begin with, and therefore no binning was necessary. Therefore, there is only one hyper-parameter in all of our work that our methodology depends on, that is the number of delays used in constructing the Hankel matrix, which we discuss in this section.

Typically, fat (more columns than rows) matrices are rank deficient and have been known to cause instabilities within DMD algorithms[51]. Additionally, applying DMD to a fat data matrix involves solving an underdetermined system which in general does not have a unique solution. Thus, one can expect that choices of $(\Delta X, \Delta T)$ leading to substantially more columns than rows $(\Delta T > \Delta X)$ can potentially be unstable for a DMD algorithm. Indeed, our choice of $\Delta X = 20$ feet and $\Delta T = 5$ s yielded a rather fat (104, 540) sized data matrix for the US-101 highway data. The reason for the stability in our results is due to the choice of delay $d = 7$ which implies that our embedded matrix is actually of size (728, 536) and hence a tall matrix (more

rows than columns). In this situation, one is solving an overdetermined system which usually has a unique solution in the least-squares sense. Furthermore, upon inspecting the eigenvalues we obtain with $d = 5$ and $d = 6$ we still find a purely real eigenvalue of $\lambda = 1$. This should not be the case since we previously subtracted the average. A delay of $d = 7$ yielded a decomposition without an eigenvalue of one while ensuring that the resulting data matrix was tall and hence our choice for analyzing the US-101 highway data. Therefore, the matrix size dilemma led to a lower bound on the number of delays to use and the appearance of the spurious eigenvalue $\lambda = 1$ guided our upper bound on the number of delays.

In general, the optimal number of delays to use is still an open topic within Koopman operator theory. However, we confirm that our procedure for selecting the number of delays always yielded a near optimal delay by plotting, on a log-log scale, the mean absolute reconstruction error (MAE) against the number of delays $d$ which can be referenced in Supplementary Fig. 28. One can immediately observe from Supplementary Fig. 28 a substantial decrease in MAE that occurs when the embedded data matrix is at least tall. Furthermore, the green dots within Supplementary Fig. 25a–c correspond to our choices of delay determined according to our systematic procedure and confirm that our procedure always yielded reasonable choices. Most importantly, our procedure for determining the number of delays does not require the costly computation of the MAE for every possible value of delay.

**Error metrics**. To quantify the performance of our method the mean absolute error (MAE), mean relative error (MRE), and root mean squared error (RMSE) are computed according to formulas (18)–(20).

$$\mathrm{MAE} = \frac{1}{N}\sum_{i,j=1}^{n,m} |\mathbf{T}_{i,j} - \mathbf{F}_{i,j}| \tag{18}$$

$$\mathrm{MRE} = \frac{1}{N}\sum_{i,j=1}^{n,m} \frac{|\mathbf{T}_{i,j} - \mathbf{F}_{i,j}|}{|\mathbf{T}_{i,j}|} \tag{19}$$

$$\mathrm{RMSE} = \sqrt{\frac{1}{N}\sum_{i,j=1}^{n,m} |\mathbf{T}_{i,j} - \mathbf{F}_{i,j}|^2} \tag{20}$$

In the above formulas, $\mathbf{T}$ is the true data matrix, $\mathbf{F}$ the forecasted data matrix, $n$ is the number of rows in $\mathbf{T}$ (number of sensor locations), $m$ the number of columns in $\mathbf{T}$ (number of time points), and $N = n \cdot m$ is the total number of elements in $\mathbf{T}$. The spatial and temporal averages of the absolute error (SMAE) and (TMAE) as well as the spatial and temporal correlations (SCorr) and (TCorr) are computed according to formulas (21) and (22). Here $\mathbf{E} = |\mathbf{T} - \mathbf{F}|$ is the absolute error matrix. The average value of correlation coefficients across different detectors (rows of $\mathbf{T}$) is used to compute what we refer to as the spatial correlation. The temporal correlation is computed by reshaping (vectorizing) $\mathbf{T}$ and $\mathbf{F}$ into a single vector time series and computing their corresponding correlations.

$$\mathrm{TMAE} = \frac{1}{m}\sum_{j=1}^{m} \mathbf{E}_{i,j}, \quad \mathrm{SMAE} = \frac{1}{n}\sum_{i=1}^{n} \mathbf{E}_{i,j} \tag{21}$$

$$\mathrm{SCorr} = \frac{1}{n}\sum_{j=1}^{m} Corr(\mathbf{T}_{i,j}, \mathbf{F}_{i,j}), \quad \mathrm{TCorr} = Corr(\mathbf{T}_{i,j}, \mathbf{F}_{i,j}) \tag{22}$$

## Data availability

The raw NGSIM trajectory data is publicly available at https://data.transportation.gov/Automobiles/Next-Generation-Simulation-NGSIM-Vehicle-Trajector/8ect-6jqj. The PeMs database can be accessed at http://pems.dot.ca.gov. All data utilized within this study is made available at https://github.com/Allan-Avila/Highway-Traffic-Dynamics-KMD-Code.git or from the authors upon request. The source data underlying Figs. 1b–d, 6a, c and Supplementary Figs. 1a–c, 2a–c, 23a, and 24a are provided as a source data file and at the above-mentioned Github repository.

## Code availability

All codes used in this study are available at: https://github.com/Allan-Avila/Highway-Traffic-Dynamics-KMD-Code.git or available from the author's upon request.

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

## Acknowledgements

This research was supported in part by the ARO-MURI Grant W911NF-14-1-0359 and the DARPA Grant HR0011-16-C-0116.

## Author contributions

A.M.A. designed, performed the research, and analyzed results. All authors were involved in discussions to interpret. I.M secured funds to support this work, and helped design the research and analysis. A.M.A. wrote the paper and all authors helped review and edit.

## Competing interests

The authors declare no competing interests.
