## [Peer Review File · Nature Communications]

Reviewers' comments:

Reviewer #1 (Remarks to the Author):

This paper builds data driven models for traffic systems based on the Koopman mode decomposition. The authors analyze the NGSIM and Caltrans highway data with the Hankel DMD algorithm. Overall the idea of using data driven analysis to study traffic dynamics is interesting and may be useful for reducing congestion. Paper introduction is clear and well-written, but after the analysis and conclusions are a bit scattered. I am a bit torn about the results of this paper. I wonder if this analysis would be more relevant for a more focused journal on traffic dynamics or data driven analysis, as this mathematical analysis may not be of great interest to the broader community. In addition, the primary results seem to indicate that even given the best-case prediction/forecasting there are large (90%) errors locally in space, which may limit the direct usefulness of this method. I am also struggling to find several relevant details about the method and parameters that will make it difficult to reproduce these results. No code or algorithms are presented to accompany the manuscript.

Specific issues:

* It is unclear from the text what the NGSIM data is actually measuring. Is it traffic density? Flow rate? Number of cars?

* I couldn't follow the use of the Hankel-DMD algorithm, which appears to be the main method. There are no details on this method at all, and the discussion of equation 2 didn't help me understand this. I don't think the reader will be able to follow what the method is actually computing from this.

* The paper does not seem to conform to reproducibility standards. Main details about the method aren't provided and there is no code accompanying the text. I strongly encourage the authors to make their code available.

* Figure 2 takes up a lot of space to show these Koopman Modes, and it is unclear what the interpretation should be. The labels seem distracting, and the red dots are not explained.

* Figure 3 plots data for 6 days, so it is impossible to see errors at this resolution. Is the entire right panel a forecast (so each 15 minutes is the forecast from the previous 15 minutes? I assume so)

** I am having a very hard time reconciling figs 3 and 4, since fig 4 says there are large regions that have ~90% error, but this doesn't seem to be reflected in fig 3. Am I missing something important?

* Figure 5 is nice, and shows that short sampling and predictions are best (very short forecasting windows). But "5%" error might not be a good agglomerate metric, considering that this "average" error is very heterogeneous in space, so that some regions are completely incorrectly predicted (90%) error, even for the best-case scenario.

Minor:

* Put acronyms in parentheses ("PLC" "MLC" etc)

* abstract: "To accurately analyzing" (for analyzing or to analyze)

* "US 101 consists of five main lanes" "US 80 consisted of six mainlines" ... be consistent here

Reviewer #2 (Remarks to the Author):

In their manuscript "Highway Traffic Dynamics: Data-Driven Analysis and Forecast", the authors present a non-parametric, "model-free" approach to characterize the dynamics of traffic on highways, specifically focusing on spatiotemporal structures of the velocity profile. The method, based on the analysis of Koopman modes, is employed to extract and identify established spatiotemporal patterns such as pinned and moving localized clusters as well as identify more exotic variants; confirm hypotheses of on/off ramp dynamics; is scalable to multi-lane traffic dynamics, and finally is used to demonstrate traffic forecasting.

While the methodology presented is impressive in its scope and application, and some interesting phenomena is uncovered, particularly the fact that traffic forecasting seems to be dependent on temporally local patterns (indicating that training forecasting algorithms on historical data might be counter-productive), I do have some concerns that preclude me from issuing a favorable recommendation to publish. I list these below:

- 1) The paper as it is presented is a bit thin on details and can benefit immensely via the inclusion of a Supplementary Information document. Given that Nat Comm is an interdisciplinary journal with a wide audience, I think it will be beneficial to not just include references to the various spatiotemporal patterns mentioned in the text, but include a synopsis of their typology as well as their significance in understanding (and mitigating) traffic congestion.
- 2) Similarly, given that it is important for the results to be reproducible, the SI should include (with examples) more details on the data and an illustration of the application of the method. Even after reading the paper three times, I was a bit confused as to what exactly constitutes the data matrix that goes into Eq. 1? I'm assuming it's the velocity vector? Please be more clear on this. Furthermore, statements such as "a customized software application was then used to transcribe the vehicle trajectory data from the video" in the caption of Figure 1 is unhelpful. Please provide references.
- 3) In the section on traffic forecasting, the authors make a case for their method being robust (~5% error when compared to the ground truth). While this is indeed impressive, no comparison is made to the relative performance of the algorithm to existing methodologies. Please consider doing so, if not in the main manuscript, then in the Supplementary Information. Also, do clarify the statement "This localized nature of the error is beneficial since localized error does not distort global structures by much" in the last para of page 11.
- 4) As I understand it, the main emphasis of the paper is to suggest KMD as an effective tool for traffic analysis, and indeed the authors have provided impressive evidence for this. I am however unclear as to the scientific advances in understanding the phenomenon of traffic that is presented in this manuscript. The authors state that in addition to existing spatiotemporal structures they uncover entirely new phenomena. Yet, it is unclear to me what these new results are. My best guess is that the authors are referring to the multi-lane analysis which is presented through a series of videos of the various Koopman modes. My concern with this analysis is that it is entirely qualitative and not quantitative. Even reading through the description of the videos, one is uncertain as to what novel spatiotemporal trends are being uncovered. This part more than anything can benefit from a clearer presentation and a more quantitative treatment.

In summary, the manuscript appears to be in essence a methods paper with the primary novelty being the domain of application. While I recognize that this is a subjective opinion, it seems to me that even the domain of application is a bit narrow in scope (especially given the fact that there appear to be no new fundamentally different insights presented on the dynamics of traffic). Consequently, in my judgment, the manuscript may be more appropriate for a specialized journal. Having said that I'm willing to hear counter-arguments from the authors.

Some minor nitpicks:

- 1) Last sentence of page 1: "There are many mathematical and artificial intelligence traffic models; all of which have had limited success at capturing certain traffic phenomena." Please provide references.
- 2) On page 7 "...Evidence for such a phenomenon can also be found by plotting the growth/decay rates (real part of eigenvalue) against the period. Both of the plots mentioned can be found in the supplemental material." Where is the supplemental material?
- 3) Page 6, 6'th line: "Which corresponds with the Koopman mode theory, where harmonics of eigenvalues are also eigenvalues (reference)." I'm assuming there is a reference that should be here?
- 4) The references are all messed up. Authors names are missing. I suspect compilation issues with the source tex file. Please fix.

This letter is point by point response to the referee's comments and summarizes the changes we have made to our previously submitted manuscript, "Highway Traffic Dynamics: Data Driven Analysis and Forecast" by Allan M. Avila and Dr. Igor Mezić.

We would first like to thank our reviewers for the positive feedback on our results:

Reviewer#1: "Overall the idea of using data driven analysis to study traffic dynamics is interesting and may be useful for reducing congestion".

Reviewer#2: "...the methodology presented is impressive in its scope and application, and some interesting phenomena is uncovered,...".

We are also thankful for further considering our work and address the following specific concerns regarding our original manuscript.

Dr. Dubrovina: You will see that, while the reviewers find your work of interest, they raise substantive concerns that cast doubt on the advance your findings represent over earlier work and the strength of the novel conclusions that can be drawn at this stage. Unfortunately, these reservations are sufficiently important to preclude publication of this study in *Nature Communications*.

We believe the manuscript has been drastically improved to additionally demonstrate the novelty and physical relevance in our work. We also improved our discussion of why they provide significant advances over previous works and added the relevant references. We emphasize that we have not altered our originally proposed algorithm or methods in any way but only extended our results by validating over a larger scope of examples such as networks and multi-lane networks. Our multi-lane network modes reveal the cyclic behaviour of five of the largest highways in Los Angeles. From these multi-lane network modes we extract how the highways operate as a network and can conclude which highways operate in phase and which carry the most magnitude of traffic and the order in which there congestion occurs. We have provided the supporting references indicating that such a successful forecasting of highway traffic, over the wide range of scenarios as those presented in this work, has never been published before.

Reviewer#1: "Paper introduction is clear and well-written, but after the analysis and conclusions are a bit scattered."

We apologize for the scattered nature of our original manuscript. We have now substantially revised and expanded our manuscript to be more understandable and have made the methodology more clear. We have also added many new figures that clearly illustrate our analysis and the results we obtain.

Reviewer#1: "I wonder if this analysis would be more relevant for a more focused journal on traffic dynamics or data driven analysis, as this mathematical analysis may not be of great interest to the broader community"

We believe these results would be of interest to a broad community since they reveal new phenomena not uncovered by other methods and indicates clear limitations on the ability to forecast traffic systems which have large societal relevance and importance. This is intimately connected with the main issue discussed within our manuscript in that the inherent limitation of many traffic forecasting algorithms is not their accuracy alone, but more so their inability to easily (without tuning/retraining) generalize to a large scale implementation across differing highways. This limitation is so severe that despite decades of research and a plethora of methodologies developed (ODE/PDE's, statistical mechanics, cellular automaton, time series, support vector regression, state estimation, clustering, Bayesian, neural networks, deep learning,...) there is still no accurate and scaleable method of forecasting and identifying spatiotemporal traffic conditions at the scale of even a single multi-lane highway, let alone multi-lane network of highways. Our method provides accurate forecasts of spatiotemporal traffic patterns at the level demanded by real world conditions.

Reviewer#1: "In addition, the primary results seem to indicate that even given the best-case prediction/forecasting there are large (90%) errors locally in space, which may limit the direct usefulness of this method."

"I am having a very hard time reconciling figs 3 and 4, since fig 4 says there are large regions that have 90% error, but this doesn't seem to be reflected in fig 3. Am I missing something important? In addition, the primary results seem to indicate that even given the best-case prediction/forecasting there are large (90%) errors locally in space, which may limit the direct usefulness of this method."

I am also struggling to find several relevant details about the method and parameters that will make it difficult to reproduce these results. No code or algorithms are presented to accompany the manuscript."

We have now polished and included the code for the most important algorithms developed. The previous results contained an overestimate of the error due to an error in data comparison. The result is actually better than stated in the previous version, and we are thankful to the referee for pointing this out. This correction now yields consistent results across the figures.

Additionally, we have computed more widely used error metrics such as the MAE, MRE and RMSE which can "loosely" be compared with other methods. We find that on average our method obtains an MAE of 1.5 (table 1 in supplemental material) in comparison to state of the art benchmarks of about 10-25 MAE. A table summarizing our error metrics is included in the supplemental information.

More importantly, we clarify in our revised manuscript that due to the presence of stochastic events (accidents and weather) within a traffic system an absolutely "perfect" forecast is impossible. Thus an equally important error metric is the difference in histograms of the real and forecasted data. We have incorporated this concept in our error analysis and conclude that indeed not only do we obtain low values of error but we also match the shape of velocity and density histograms of the original data. This indicates that our methodology can indeed match the statistics of the actual system despite the existence of stochastic events within traffic systems.

Reviewer#1: It is unclear from the text what the NGSIM data is actually measuring. Is it traffic density? Flow rate? Number of cars?

We have now included an additional figure (figure 1) that makes this clear. We have also expanded our description of the NGSIM data set and how we utilize it to obtain spatiotemporal profiles of velocity, density and flow. We take this opportunity to clarify that our conversion of the NGSIM data from vehicle trajectory data to spatiotemporal data is a consequence of the format in which the data was acquired by the federal highway administration and our analysis algorithms are independent of this. For example, the PeMs data collected by the California department of transportation provides spatiotemporal profiles of velocity, density and flow and therefore we did not need to convert it. Lastly, our entire research was focused on the macroscopic analysis and forecasting of traffic systems and not on the microscopic description of individual vehicles hence our choice of working with spatiotemporal data rather than the raw NGSIM data. We have also provided references for other works which have similarly converted the NGSIM data in their analysis.

Reviewer#1: "I couldn't follow the use of the Hankel-DMD algorithm, which appears to be the main method. There are no details on this method at all, and the discussion of equation 2 didn't help me understand this. I don't think the reader will be able to follow what the method is actually computing from this."

Reviewer#1: "The paper does not seem to conform to reproducibility standards. Main details about the method aren't provided and there is no code accompanying the text. I strongly encourage the authors to make their code available."

We have now included an additional figure (figure 2) that clearly describes what the Hankel-DMD algorithm is computing. Additionally, we have included in the supplemental information an expanded discussion on the details of the Hankel-DMD algorithm and its implementation. We have also provided the source code and algorithm for the Hankel-DMD.

Reviewer#1: "Figure 2 takes up a lot of space to show these Koopman Modes, and it is unclear what the interpretation should be. The labels seem distracting, and the red dots are not explained."

We have expanded the description of the Koopman modes within both the main text and the figure description. We reference how they correspond to previously observed phenomena as well as discover never before identified patterns. The red dots were explained to be the locations of the on and off-ramps however, in case this was not sufficiently clear we have now repeated this explanation several times within both the main text and figure description.

Reviewer#1: "Figure 3 plots data for 6 days, so it is impossible to see errors at this resolution. Is the entire right panel a forecast(so each 15 minutes is the forecast from the previous 15 minutes? I assume so)" We have relabeled our figures to more clearly distinguish the raw and forecasted data. We have additionally specified that the last 15 minutes were used to forecast the next 15 minutes of data within the figure description.

Reviewer#1: "Figure 5 is nice, and shows that short sampling and predictions are best (very short forecasting windows). But "5%" error might not be a good agglomerate metric, considering that this "average" error is very heterogeneous in space, so that some regions are completely incorrectly predicted (90%) error, even for the best-case scenario. "

Thank you for the positive remarks on our results. Once again this discrepancy was due to a bug in our computation of the spatial error. In the process of polishing the code for publication we discovered and corrected this bug and the results are now consistent and the error estimate is in fact lower.

Additionally, we have expanded our forecasting results to the multi-lane network scenario and have provided a movie of the result. Within the movie we plot the forecasts of the next fifteen minutes along with the real data lagged by fifteen minutes so that one can see the forecasts and then visually compare with the what actually happened. The video demonstrates that the traffic patterns forecasted actually occur within the true data set fifteen minutes later and allows one to visually verify that our forecasted

patterns are actually occurring in the future.

Reviewer#1 Minor edits:

(1) "Put acronyms in parentheses ("PLC" "MLC" etc)"

(2) "abstract: "To accurately analyzing" (for analyzing or to analyze)"

(3) "US 101 consists of five main lanes" "US 80 consisted of six mainlines" ... be consistent here"

(1) We currently place acronyms within parenthesis only the first time they are defined and not at every subsequent use. If this is incorrect grammar we are more than willing to always place them in parenthesis.

(2) This has been corrected.

(3) This has been corrected.

Reviewer#2: "The paper as it is presented is a bit thin on details and can benefit immensely via the inclusion of a Supplementary Information document. Given that Nat Comm is an interdisciplinary journal with a wide audience, I think it will be beneficial to not just include references to the various spatiotemporal patterns mentioned in the text, but include a synopsis of their typology as well as their significance in understanding (and mitigating) traffic congestion. "

The supplementary material was included in the main text after the bibliography. We did this for ease of reference and apologize that this made it difficult for you to locate. We have expanded our supplemental information and attached it as a separate document. We are also more than willing to add to it as the reviewers deem necessary. We have also included a discussion within the main text on how the ability to identify spatiotemporal patterns can be coupled with ramp-metering algorithms to aid in the control of traffic congestion and increase safety. The corresponding references are included.

Reviewer#2: "Similarly, given that it is important for the results to be reproducible, the SI should include (with examples) more details on the data and an illustration of the application of the method. Even after reading the paper three times, I was a bit confused as to what exactly constitutes the data matrix that goes into Eq. 1? I'm assuming it's the velocity vector? Please be more clear on this. Furthermore, statements such as "a customized software application was then used to transcribe the vehicle trajectory data from the video" in the caption of Figure 1 is unhelpful. Please provide references. "

We have expanded our description of the data sets and clearly state what constitutes a data vector. We have also expanded our discussion of the Hankel-DMD algorithm both within the text and the supplemental information. We have also included two additional figures (figure 1 and 2) illustrating how the data is obtained and a flow scheme of how the data is decomposed via the Hankel-DMD. The algorithm and source code have also been included in the supplemental information.

Reviewer#2: "In the section on traffic forecasting, the authors make a case for their method being robust (5% error when compared to the ground truth). While this is indeed impressive, no comparison is made to the relative performance of the algorithm to existing methodologies. Please consider doing so, if not in the main manuscript, then in the Supplementary Information. Also, do clarify the statement "This localized nature of the error is beneficial since localized error does not distort global structures by much" in the last para of page 11."

In an effort to be able to compare to existing methodologies we have implemented the more widely used mean absolute error as opposed to a % error. In doing so, we find that a direct comparison between our works and others is nearly impossible. This is due to the well known fact that comparing the performance of differing methodologies can be difficult as their respective objectives and methods may vastly differ. For example, many of the existent state of the art benchmarks were obtained over very controlled (filtering/smoothing/removing seasonal averages) and limited data (weekday/weekend/holiday/few detectors), and under extensive training and tuning of model parameters. On the other hand, our methodology does not rely on any filtering, pre-processing, distinguishing of weekday, weekend, holiday data nor the removal of seasonal averages to function accurately. Our method is verified and stable over hundreds of miles of highway detectors and across holiday and adverse weather conditions. Our method does not rely on any large historical training data nor the tuning of parameters or prior knowledge of any model. For these reasons, we focus mainly on contrasting, as opposed to comparing, our works to prior literature. Nevertheless, we draw loose comparisons to other methods and find that existing methodologies perform within the range of 15-30 mean absolute error over very limited data and extensive training. Our Method, on average performs between 1.5-8 mean absolute error.

We agree that the statement "This localized nature of the error is beneficial since localized error does not distort global structures by much" is very vague and has been removed from the manuscript.

Reviewer#2: "As I understand it, the main emphasis of the paper is to suggest KMD as an effective tool for traffic analysis, and indeed the authors have provided impressive evidence for this. I am however unclear as to the scientific advances in understanding the phenomenon of traffic that is presented in this manuscript. The authors state that in addition to existing spatiotemporal structures they uncover entirely new phenomena. Yet, it is unclear to me what these new results are. My best guess is that the authors are referring to the multi-lane analysis which is presented through a series of videos of the various Koopman modes. My concern with this analysis is that it is entirely qualitative and not quantitative. Even reading through the description of the videos, one is uncertain as to what novel spatiotemporal trends are being uncovered. This part more than anything can benefit from a clearer presentation and a more quantitative treatment. "

We first thank you for stating our results as being impressive.

We have revised our manuscript to more clearly describe the novel phenomena we uncover. For example, we explain how the pumping effect modes we identify display evidence for the previously proposed phenomena that were never identified from data. We have extended our description of the the multi-lane modes to describe how they also uncover the previously postulated effects of lane changing behaviour on traffic systems. Specifically, the zig-zag modes we uncover display the lateral, across lane, wavelike dynamics that lane changing maneuvers can induce as proposed but never identified with such coherent patterns. We have also described how the the multi-lane modes can be used to quantify the performance of on-ramp metering and can aid in the control of on-ramp traffic. We also utilize the eigenvalues to provide evidence of previously postulated phenomena regarding the persistence of slower frequency modes, the relation between their average amplitudes and demonstrate the existence of fundamental frequencies that are common across differing observables (velocity, density and flow).

The data-driven analysis we have employed is firmly rooted in operator theoretic methods in dynamical systems. Specifically, eigenvalues represent the quantitative time scales and modes give the quantitative description of the spatial structure of the associated traffic, see equation 2 in our manuscript.

Reviewer#2: "In summary, the manuscript appears to be in essence a methods paper with the primary novelty being the domain of application. While I recognize that this is a subjective opinion, it seems to me that even the domain of application is a bit narrow in scope (especially given the fact that there appear to be no new fundamentally different insights presented on the dynamics of traffic). Consequently, in my judgment, the manuscript may be more appropriate for a specialized journal. Having said that I'm willing to hear counter-arguments from the authors. "

The insights that our results provide into traffic dynamics begin with extracting empirically observed phenomena directly from data. Thus verifying that these previously identified patterns in fact exist within traffic systems and can be measured directly from data. Our results also provided insight into previously postulated but unobserved phenomena such as the "pumping effect", multi-lane zig-zag patterns induced by lane changing behaviour as well as multi-lane MLC patterns. Additionally, our multi-lane network analysis reveals detailed insight into the traffic dynamics of networks. For example, our results demonstrate which highways operate in phase and which carry the most magnitude of traffic. our results give insight into the fact that highway networks congest in a certain order and our identified patterns reveal the order of congestion. Lastly, all of the patterns we extract carry an exact physical meaning and have an associated timescale which provides insight into the frequency of specific patterns within traffic dynamics.

We now argue that the forecasting methodology we have developed is in fact quite general and can be applied across different fields of study beyond highway traffic. For example, researchers in the field of finance utilize time series methods such as auto-regressive integrated moving average (ARIMA) models along with generalized auto-regressive conditional heteroskedasticity (GARCH) models to forecast the average and fluctuating components of economic systems respectively. In the case of a purely periodic system the standard approach is to utilize Fourier transform methods. In general one can attempted to model traffic or some complex system as a sum of three components corresponding to an average trend, a periodic component and a fluctuating or unpredictable component. While this is indeed the triple decomposition contained within the spectrum of the Koopman operator, historically, researchers have not been able to accurately extract these components. Our methodology demonstrates how the Koopman mode decomposition yields the correct decomposition of the system and can be utilized to forecast complex systems. We are convinced that our results will be of interest to a community much broader than that of a specialized traffic analysis journal due to the multidisciplinary nature of forecasting complex systems such as traffic, finance and disease modeling, to name a few, which are all of large societal impact and relevance.

Reviewer#2 Minor edits:

- (1) "Last sentence of page 1: "There are many mathematical and artificial intelligence traffic models; all of which have had limited success at capturing certain traffic phenomena." Please provide references."
- (2) "On page 7 "...Evidence for such a phenomenon can also be found by plotting the growth/decay rates (real part of eigenvalue) against the period. Both of the plots mentioned can be found in the supplemental material." Where is the supplemental material? "
- (3) "Page 6, 6'th line: "Which corresponds with the Koopman mode theory, where harmonics of eigenvalues are also eigenvalues (reference)." I'm assuming there is a reference that should be here?"
- (4) "The references are all messed up. Authors names are missing. I suspect compilation issues with the source tex file. Please fix."

- (1) Sentence has been revised and the corresponding references included.
- (2) Supplemental material was included after the bibliography. We have now included it as a separate document for your ease of referencing and once again apologize that you were unable to originally find it.
- (3) The appropriate reference has been included.
- (4) This has been fixed and the number of references has been incremented.

We now conclude, without any reference to a specific reviewer's comment, we are more than willing to reorganize the figures/content in any way and extend the currently submitted supplemental material to include any material we might still be lacking. Thank you, once again, for your valuable revisions and further considering our work and we thank you again for your feedback as it has led to many improvements of the manuscript.

Sincerely,

Dr. Igor Mezić & Allan M. Avila

Reviewers' comments:

Reviewer #1 (Remarks to the Author):

I want to thank the authors for their revised manuscript. The reconstruction and forecasting performance appears to be good (although MAE should be replaced with either a normalized error throughout or given meaningful units). But I am still struggling with many same problems with the original. Also the changes in the revision are not colored so I can't tell what has changed without cross reference. A big concern is that there is no comparison with any other method, and there is the strong claim that "the forecasting scheme we propose vastly outperforms many state-of-the-art benchmarks".

I am still concerned that this paper seems like a focused application of a mathematical method to a specific problem. It is not clear what general knowledge is gained that can be used in other fields.

The graphics in this paper are not up to the standard of this journal, which generally has very clear and intuitive figures to illustrate ideas. Some examples:

— figure 1: many fonts are tiny and illegible; graphics are pixelated and panel (c) looks distorted.

The acronyms throughout are not defined in caption

— figure 2: very unclear and disorganized graphic. Text is stretched, figures are not labeled and don't have axes; what do letters mean? Why not use standard notation of data in time $x(t)$ or x_k from dmd

— figure 3: tiny labels, completely illegible, even when printed; no axes labels; the circles and squares and colors are not defined in the caption; these labels are mentioned in the main text but don't appear to mean anything

— Results in fig 7 are for 15 min intervals, but results are so pixelated and interpolated it is impossible to actually see these results.

— Labels in fig 7 also wrong. X-axis goes from 0-168 on top, 1st-30th on middle and 22nd -26th on bottom. Left can't all be "hours" and middle can't all be "date".

— Figure 8: INSTEAD OF MAE, shouldn't plot show MRE (eq 4). Also, eq 4 is wrong! needs norm on T

— figure 10: gray background not helpful; fonts all stretched and illegible, many plots with no axes; some legends/colorbars are cutoff

I don't think these figures convey the information in a way that will be easy to comprehend. The reader has to work far too hard to understand what is the content.

There are a number of important questions that also must be asked for this study to really be useful for realtime traffic:

* How are the parameters chosen for this example; how would they be chosen in practice on a new data set

There are a number of statements in the introduction that appear vague and unsubstantiated:

* "This number is higher for some European countries " should have a reference

* "Further- more, the forecasting scheme we propose vastly outperforms many state-of-the-art benchmarks " needs to be substantiated. In fact, I see no comparison with state of the art. I don't see how the paper can be published without a clear and prominent comparison.

In revised paper please color code changes.

Minor issues

"Has enabled enormous amount of" is not proper English

Sentence before equation (2) should not have a period ; also math does not look good. Why not use latex; equation 2 should not be punctuated with comma

Also, eq 4 is wrong. needs norm on T in denominator

Reviewer #2 (Remarks to the Author):

The new version of the manuscript represents a significant improvement over the previous edition, and the authors are thanked for attending to the suggestions in the right spirit. The manuscript is now coherent, well-motivated, and a good job has been done in both enabling reproducibility of the results as well as putting their novelty in context.

A minor nitpick is in terms of how the SI is organized and referenced in the main manuscript. Too often the authors just refer to supplemental information, rather than the specific section, figure or equation, which made it a bit difficult to navigate the paper.

Pending the correction of this issue of readability, I have no objection to publication.

Reviewer#1:"MAE should be replaced with either a normalized error throughout or given meaningful units"

Thank you for pointing this out to us. Units have been assigned to the MAE.

Reviewer#1:"Changes in the revision are not colored so I can't tell what has changed without cross reference."

We point out that our previous manuscript was over the 5000-word limit and therefore many sections have been shortened or rearranged into the methods/supplemental information. For this reason, we believe we would essentially color code the entire document, however, we understand this is not useful in any way to our reviewers. We have therefore included a copy of the previous manuscript with appropriate commentary indicating how the manuscript has changed. Please refer to the document "Old Manuscript with Highlighted revisions" for the details of how the paper was revised and reorganized.

Reviewer#1:"A big concern is that there is no comparison with any other method, and there is the strong claim that "the forecasting scheme we propose vastly outperforms many state-of-the-art benchmarks"."

Thank you for the suggestion, we have decided to remove the claim. Namely, it is known that a completely objective comparison of traffic forecasting methodologies is difficult. For example, differing studies utilize different highways and data-sets, methodologies differ in the amount of filtering, pre-processing, training data set size and forecast horizons. The few comparisons of traffic forecasting methods and models that exist are often times complete state of the art review papers which take great care to obtain an objective as possible comparison. For these reasons in addition to our paper not being a focused comparison of methodologies, we found it is best to remove the claim.

Reviewer#1:"I am still concerned that this paper seems like a focused application of a mathematical method to a specific problem. It is not clear what general knowledge is gained that can be used in other fields."

We point out that our methodology makes absolutely no assumptions that the underlying system is highway traffic. Specifically, we only assume to have time ordered data arising from observations of a linear or nonlinear dynamical system. With that arbitrary data we are able to identify, analyze and forecast important sub patterns within the data and relate them to physically meaningful phenomena. Therefore we argue that the forecasting methodology we have developed is in fact quite general and can be applied across different fields of study beyond highway traffic. For example, researchers in the field of finance utilize time series methods such as auto-regressive integrated moving average (ARIMA) models along with generalized auto-regressive conditional heteroskedasticity (GARCH) models to forecast the average and fluctuating components of economic systems respectively. In the case of a purely periodic system, the standard approach is to utilize Fourier transform methods. In general, one can attempt to model traffic or some complex system as a sum of three components corresponding to an average trend, a periodic component and a fluctuating or unpredictable component. While this is indeed the triple decomposition contained within the spectrum of the Koopman operator, historically, researchers have not been able to accurately extract these components. Our methodology demonstrates how the Koopman mode decomposition yields the correct decomposition of the system and can be utilized to forecast complex systems. We are convinced that our results will be of interest to a community much broader than that of a specialized traffic analysis journal due to the multidisciplinary nature of forecasting complex systems such as traffic, finance and disease modeling, to name a few, which are all of large societal impact and relevance.

Reviewer#1: "The graphics in this paper are not up to the standard of this journal, which generally has very clear and intuitive figures to illustrate ideas. Some examples: "

1. "figure 1: many fonts are tiny and illegible; graphics are pixelated and panel (c) looks distorted. The acronyms throughout are not defined in caption"
 2. "figure 2: very unclear and disorganized graphic. Text is stretched, figures are not labeled and don't have axes; what do letters mean? Why not use standard notation of data in time $x(t)$ or x_k from dmd"
 3. "figure 3: tiny labels, completely illegible, even when printed; no axes labels; the circles and squares and colors are not defined in the caption; these labels are mentioned in the main text but don't appear to mean anything"
 4. "Results in fig 7 are for 15 min intervals, but results are so pixelated and interpolated it is impossible to actually see these results."
 5. "Labels in fig 7 also wrong. X-axis goes from 0-168 on top, 1st-30th on middle and 22nd -26th on bottom. Left can't all be "hours" and middle can't all be "date"."
 6. "Figure 8: INSTEAD OF MAE, shouldn't plot show MRE (eq 4). Also, eq 4 is wrong! needs norm on T"
 7. "figure 10: gray background not helpful; fonts all stretched and illegible, many plots with no axes; some legends/colorbars are cutoff"
1. We have relocated this figure to the supplemental information in addition to dividing it into two separate images for a less pixelated display. In its place, we have decided to only demonstrate the resulting spatiotemporal data set as they are the relevant displays. We have taken care to make all fonts large and visible.
 2. We have also relocated this figure to the supplemental information as it is not relevant to the results but more so to the methods. In doing so we have also implemented standard notation as you suggest. However, we have not labeled the individual figures since this is a flow scheme of a method rather than a panel of figures. A figure similar to this was published in the HAVOK nature communications paper which we have been asked to cite. There are no relevant axes here since this is a flow scheme so none have been added. We have also taken care to reduce the stretching of the text.
 3. This figure has also been divided into two figures so that all labels and text is more visible. The axes have now been labeled, and the colored circles and squares have been removed.
 4. The results of this figure are for months and weeks worth of data with a 15-minute resolution and it is therefore difficult to capture all the fine details. For this reason, we have removed the month's data and relocated it as a separate figure in the supplemental information. This has allowed us to enlarge the other remaining figures for a less pixelated figure.
 5. Thank you for pointing this out to us the labels have now been corrected, made legible and each panel contains its own labels.
 6. We understand that MRE is a typical error metric within data analysis. However, we have chosen to use MAE since it carries physically meaningful units, which have now been assigned to the figures. We believe reporting an error of 2-3 miles per hour is perhaps more meaningful than a percent. Within equation (4) we meant the element wise division, we have updated our notation to reflect this and have added absolute value bar symbols on the denominator.
 7. This figure has now been split into two figures so as to enlarge the images and make them much more legible. The grey background has been removed, every plot has been given axes and labels and color-bars are no longer cut off.

Reviewer#1: "I don't think these figures convey the information in a way that will be easy to comprehend. The reader has to work far too hard to understand what is the content."

Thank you for providing such a detailed review of our figures. We have incorporated all of your suggestions and believe that the figures are now much more readable and understandable.

Reviewer#1: "There are a number of important questions that also must be asked for this study to really be useful for realtime traffic:"

1. "How are the parameters chosen for this example; how would they be chosen in practice on a new data set"
1. This was explained in the supplemental information. We have now moved that relevant section of the supplemental information to the methods section. Furthermore, we would like to point out that the spatiotemporal bin size ΔX and ΔT used to convert the Lagrangian NGSIM data into Eulerian data is not relevant to our methodology but specific to how the data was presented. For example, the Caltrans PeMs data was provided as Eulerian data, to begin with, and therefore no binning was necessary. Therefore, there is only one hyper-parameter in all of our work that our methodology depends on, that is the number of delays used in constructing the Hankel matrix, which we have explained in detail how to select in our previous supplemental information which is now part of our methods sections.

Reviewer#1: "There are a number of statements in the introduction that appear vague and unsubstantiated:"

1. "This number is higher for some European countries " should have a reference"
2. "Furthermore, the forecasting scheme we propose vastly outperforms many state-of-the-art benchmarks " needs to be substantiated. In fact, I see no comparison with state of the art. I don't see how the paper can be published without a clear and prominent comparison."
1. Thank you for pointing this out to us, the citation has been included.
2. We have previously answered this twice in the above comments. Nevertheless, our response is as follows:

Thank you for the suggestion, we have decided to remove the claim. Namely, it is known that a completely objective comparison of traffic forecasting methodologies is difficult. For example differing studies utilize different highways and data-sets, methodologies differ in the amount of filtering, pre-processing, training data set size and forecast horizons. The few comparisons of traffic forecasting methods and models that exist are often times complete state of the art review papers which take great care to obtain an objective as possible comparison. For these reasons in addition to our paper not being a focused comparison of methodologies, we found it is best to remove the claim.

Reviewer#1: "In revised paper please color code changes."

We point out that our previous manuscript was over the 5000 word limit and therefore many sections have been shortened or rearranged into the methods/supplemental information. For this reason, we believe we would essentially color code the entire document, however, we understand this is not useful in any way to our reviewers. We have therefore included a copy of the previous manuscript with appropriate commentary indicating how the manuscript has changed. Please refer to the document "Old Manuscript with Highlighted revisions" for the details of how the paper was revised and reorganized.

Reviewer#1 Minor edits:

1. "Has enabled enormous amount of" is not proper English"
2. "Sentence before equation (2) should not have a period ; also math does not look good. Why not use latex; equation 2 should not be punctuated with comma"
3. "Also, eq 4 is wrong. needs norm on T in denominator"

1. Thank you for pointing this out. The sentence has been fixed.
2. Thank you for pointing this out. We have revised our punctuation of our mathematics.
3. Within equation (4) we meant the element wise division, we have updated our notation to reflect this and have added absolute value bar symbols on the denominator.

Reviewer#2 Minor edits:

1. "A minor nitpick is in terms of how the SI is organized and referenced in the main manuscript. Too often the authors just refer to supplemental information, rather than the specific section, figure or equation, which made it a bit difficult to navigate the paper."
1. Thank you for pointing this out. All supplemental references within the main text now reference specific figures, tables etc.

We thank our reviewers and the editor, once again, for their valuable feedback as it has lead to many improvements of the manuscript.

Sincerely,

Dr. Igor Mezić & Allan M. Avila